

# Inhomogeneous quantum quenches in the sine-Gordon theory

Dávid X. Horváth[1*], Spyros Sotiriadis[2,3‡] Márton Kormos[4,5,6†] and Gábor Takács[4,5,6∘]

**1** Scuola Internazionale Superiore di Studi Avanzati (SISSA) and
INFN Sezione di Trieste, Trieste, Italy
**2** Dahlem Center for Complex Quantum Systems, Freie Universität Berlin, Berlin, Germany
**3** Faculty of Mathematics and Physics, University of Ljubljana, Ljubljana, Slovenia
**4** Department of Theoretical Physics, Institute of Physics, Budapest University of Technology
and Economics, Műegyetem rkp. 3., H-1111 Budapest, Hungary
**5** MTA-BME Quantum Dynamics and Correlations Research Group, Műegyetem rkp. 3.,
H-1111 Budapest, Hungary
**6** BME-MTA Statistical Field Theory 'Lendület' Research Group, Műegyetem rkp. 3.,
H-1111 Budapest, Hungary

⋆ esoxluciuslinne@gmail.com, † spyros.sotiriadis@fu-berlin.de,
‡ kormosmarton@gmail.com, ∘ takacs.gabor@ttk.bme.hu

## Abstract

We study inhomogeneous quantum quenches in the attractive regime of the sine–Gordon model. In our protocol, the system is prepared in an inhomogeneous initial state in finite volume by coupling the topological charge density operator to a Gaussian external field. After switching off the external field, the subsequent time evolution is governed by the homogeneous sine–Gordon Hamiltonian. Varying either the interaction strength of the sine–Gordon model or the amplitude of the external source field, an interesting transition is observed in the expectation value of the soliton density. This affects both the initial profile of the density and its time evolution and can be summarised as a steep transition between behaviours reminiscent of the Klein–Gordon, and the free massive Dirac fermion theory with initial external fields of high enough magnitude. The transition in the initial state is also displayed by the classical sine–Gordon theory and hence can be understood by semi-classical considerations in terms of the presence of small amplitude field configurations and the appearance of soliton excitations, which are naturally associated with bosonic and fermionic excitations on the quantum level, respectively. Features of the quantum dynamics are also consistent with this correspondence and comparing them to the classical evolution of the density profile reveals that quantum effects become markedly pronounced during the time evolution. These results suggest a crossover between the dominance of bosonic and fermionic degrees of freedom whose precise identification in terms of the fundamental particle excitations can be rather nontrivial. Nevertheless, their interplay is expected to influence the sine–Gordon dynamics in arbitrary inhomogeneous settings.

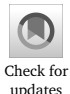

## Contents

## 1 Introduction

Non-equilibrium dynamics of quantum many-body systems have been at the forefront of research in recent years [1–4]. Among other problems belonging to this class, the dynamics induced by an initially localised disturbance in an otherwise uniform quantum fluid is a fundamental theoretical problem [5–14] which is relevant in a variety of different contexts, ranging from transport in condensed matter and atomic gases to high energy processes and early universe dynamics. Unlike the simple quantum mechanical exercise of an initially localised wave packet that is spreading with time, its many-body analogue is far more complex, especially when the background fluid on top of which the disturbance is applied is described

by strongly interacting and topologically nontrivial quantum fields. As it happens this is also the most significant from an application point of view case, which is why several trailblazing ideas have been proposed to solve variants of this problem, e.g. by means of atomic quantum simulators [5–9], tensor network based numerical methods [10–12], quantum annealer platforms [13], and ultimately quantum computers [14]. However, most solution approaches are based on a discretisation of space and time, which does not always provide a faithful approximation of the continuous nature of quantum fields.

Applying a localised initial disturbance on a quantum system can be thought of as a special instance of a quantum quench [15], a paradigmatic out-of-equilibrium protocol with great experimental relevance, and, more specifically, a quench under inhomogeneous settings [16, 17]. After reaching local equilibrium, the large scale relaxation of the system is expected to be described by hydrodynamics [18] which can be rigorously established in simple systems [19,20]. In one spatial dimension, the non-equilibrium dynamics of integrable models displays special transport properties due to the presence of higher conserved quantities [21–26], and in the hydrodynamic limit are described by the recently developed Generalised Hydrodynamics (GHD) that captures the characteristic ballistic transport [27–42].

Before reaching the regime where the hydrodynamic description applies, however, the time evolution of the system is to be described at the quantum level. This is a much harder problem which can be addressed with exact analytic methods for non-interacting systems [43–45]; for interacting integrable systems the tools available so far consist of lattice based numerical methods [21], mean field approaches [17] or, more recently, semi-classical methods [46, 47].

In this work we address the full quantum time evolution of an interacting integrable quantum field theory, the sine–Gordon model starting from inhomogeneous initial conditions. The sine–Gordon model is not merely a paradigmatic example of integrable quantum field theory [48], but it also has a wide range of applications to the description of condensed matter systems [49]. In particular, sine–Gordon field theory is expected to describe the dynamics of an extended bosonic Josephson junction formed by coupled ultra-cold one-dimensional condensates [50], including non-equilibrium processes [51]. Realising coupled bosonic condensates on an atom chip has confirmed that higher order equilibrium correlations are indeed described by the sine–Gordon model [52], and the non-equilibrium time evolution of the coupled condensates is a matter of active investigations both experimentally and theoretically [53–62]. Recently it was proposed that the model can also be realised using superconducting quantum circuits [63]. The inhomogeneous initial condition considered here is realised by coupling the soliton charge density to a position dependent external source. The case with a constant external source, i.e. a chemical potential, describes the commensurate-incommensurate phase transition and was considered in the seminal papers [64–68].

Our method of choice to investigate the dynamics is a variant of Hamiltonian truncation, the so-called truncated conformal space approach (TCSA) introduced in [69] to describe the finite volume spectrum of perturbed minimal conformal field theories, and extended to the sine–Gordon model in [70, 71]. More recently, Hamiltonian truncation approaches were applied to homogeneous quantum quenches [72–74], including those in sine–Gordon quantum field theory [58, 75, 76]. This method has the advantage that it works directly in the continuum limit so no space-time discretisation is required. Here we extend this method to the inhomogeneous case, and use it both for constructing the initial state and study its subsequent time evolution. We compare the results of these studies to classical time evolution as well as to exact analytic calculations at the free fermion point.

The central finding of this work is demonstrated by Fig. 1.1 which displays the time evolution of the topological charge density. Controlling the magnitude of the initial inhomogeneity and the intrinsic interaction strength of the sine–Gordon theory, a transition can be observed between dynamics reminiscent of the massive Klein–Gordon (free boson) theory and

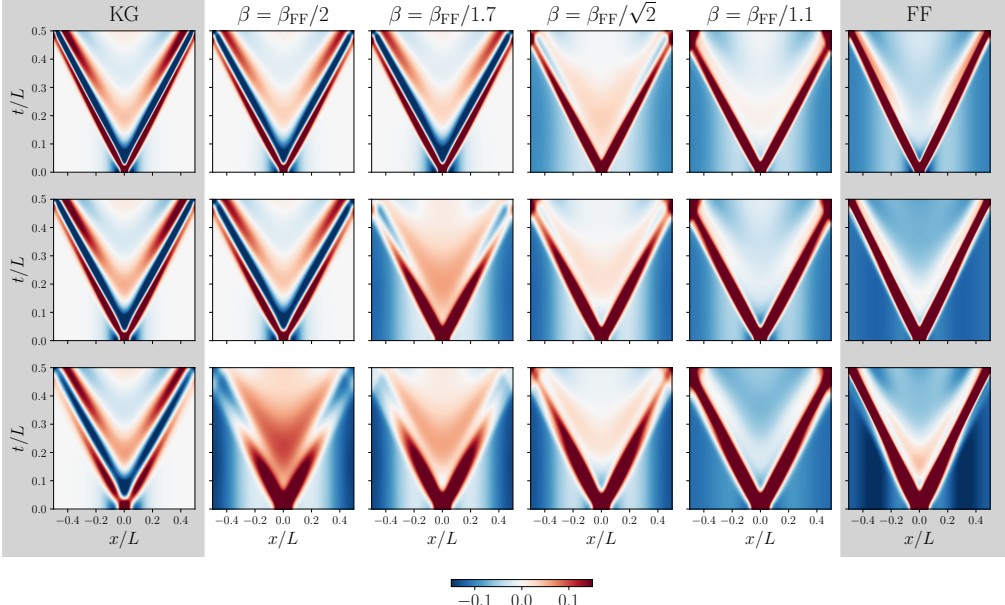

Figure 1.1: **Transition from free boson-like to free fermion-like behaviour after an inhomogeneous quantum quench in the sine–Gordon model:** Density plots of the time evolution of the soliton density $\rho(x,t) = \beta \partial_x \varphi(x,t)/(2\pi)$ as a function of the space and time coordinates $x, t$ at various values of the parameters. From left to right the sine–Gordon coupling $\beta$ changes from 0 (Klein–Gordon limit) to $\beta_{\mathrm{FF}} = \sqrt{4\pi}$ (free fermion point), while from top to bottom the initial external field bump changes from shorter and thinner to taller and wider bumps (the three rows correspond to the bump height parameter $A\beta_{\mathrm{FF}}/m_1$ and the Gaussian width parameter $m_1 \sigma$ taking the values $(12, 1/3), (15, 4/9)$ and $(18, 2/3)$, respectively). In the Klein-Gordon case ($\beta = 0$) the relation $\rho(x,t) = \beta \partial_x \varphi(x,t)/(2\pi)$ would lead to zero result, so instead we defined $\rho(x,t) = \partial_x \varphi(x,t)/(2\sqrt{\pi})$ which is the same relation which was used at the next coupling $\beta = \beta_{\mathrm{FF}}/2$, to facilitate comparison. Note the change in the propagation fronts from having oscillatory tails (Klein–Gordon dynamics) to fast decaying tails (free fermion dynamics). In addition, the background is neutral (white) in the free boson limit, while at the free fermion point a negatively charged (blue) background appears to compensate for the positive (red) bump charge. This change is triggered by a crossover transition in the initial state, which occurs at an intermediate value of $\beta$ which depends on the initial bump strength.

the massive Dirac (free fermion) theory. In this work we analyse this behaviour in detail. Besides the study of the time evolution comprising the case of the free theories, we carefully investigate the inhomogeneous initial states which also display the same transition. Their study provides a semi-classical understanding of the phenomenon by means of the classical version of the sine–Gordon model. Based on our investigations, a natural interpretation for our observation is given by the interplay between bosonic and fermionic degrees of freedom present in the quantum sine–Gordon model.

The outline of the paper is as follows. In Section 2 we introduce the setup of the system, including the initial state and the time evolution. Section 3 describes the application of the TCSA to the study of inhomogeneous quenches in the sine–Gordon model. The results of our simulations are summarised and interpreted in Section 4, and the conclusions, including details on the feasibility of an experimental observation, are presented in Section 5. In order to keep the main text focused, technical details are relegated to the Appendices. The TCSA

method and its application is discussed in Appendix A, details about the inhomogeneous initial state can be found in Appendix B, while the tools necessary to compute the free boson and fermion dynamics are summarised in Appendix C.

# 2 Sine–Gordon model and time-evolution from an inhomogeneous initial condition

## 2.1 Generalities

The sine–Gordon quantum field theory (QFT) is defined by the following action

$$\mathcal{A} = \int d^2x \left( \frac{1}{2} \partial_\mu \varphi \, \partial^\mu \varphi + \lambda \cos \beta \varphi \right), \tag{2.1}$$

where $\lambda$ is the coupling constant and $\varphi$ is a compactified scalar field with compactification radius $R = 2\pi/\beta$, i.e. $\varphi \sim \varphi + 2\pi/\beta$. The particle spectrum of the theory consists of, first of all, a soliton $s$ and an anti-soliton $\bar{s}$ forming a particle doublet and possessing opposite $U(1)$ topological charge. The topological charge density is given by

$$\rho(x,t) = \frac{\beta}{2\pi} \partial_x \varphi(x,t), \tag{2.2}$$

and the charge itself

$$Q = \int dx \, \rho(x,t) \tag{2.3}$$

corresponds to the number of solitons minus the anti-solitons. When the field is compactified on a spatial circle of circumference $L$, $Q$ is eventually the winding number of the field.

Depending on the value of the $\beta$ parameter, we can distinguish the attractive and repulsive regime of the QFT. When $4\pi \leq \beta^2 < 8\pi$, solitons and anti-solitons repel each other, whereas if $\beta^2 < 4\pi$, they attract each other and consequently can form bound states. These bound states are called breathers and are topologically neutral particles. Defining the new coupling strength

$$\xi = \frac{\beta^2}{8\pi - \beta^2}, \tag{2.4}$$

the number of different breather species is $\lfloor 1/\xi \rfloor$, where $\lfloor \bullet \rfloor$ denotes the integer part.

The sine–Gordon model is equivalent to the massive Thirring model of interacting fermions [77]

$$\mathcal{A}_F = \int d^2x \left( i\bar{\psi} \, \partial\!\!\!/ \, \psi - M\bar{\psi}\psi - \frac{g}{2}(\bar{\psi}\gamma^\mu\psi)(\bar{\psi}\gamma_\mu\psi) \right), \tag{2.5}$$

with the couplings related as

$$\frac{\beta^2}{4\pi} = \frac{1}{1 + g/\pi}, \tag{2.6}$$

and the solitons and anti-solitons corresponding to the fundamental fermionic particle and its antiparticle of the Dirac theory [78]. The bosonic coupling $\beta^2 = 4\pi$ corresponds to free dynamics $g = 0$ in terms of the fermions. For later convenience, we introduce the notation

$$\beta_{\text{FF}} = \sqrt{4\pi}, \tag{2.7}$$

and from now on we write specific values of the coupling $\beta$ relative to the above free fermion value, which also makes it much easier to compare with other conventions used in different applications of sine–Gordon theory.

The soliton mass $M$ is related to the coupling $\lambda$ and $\beta$ through the mass-coupling relation [79],

$$\lambda = \frac{\Gamma\left(\frac{\beta^2}{8\pi}\right)}{\pi\Gamma\left(1-\frac{\beta^2}{8\pi}\right)}\left[\frac{M\sqrt{\pi}\Gamma\left(\frac{4\pi}{8\pi-\beta^2}\right)}{2\Gamma(\frac{\beta^2/2}{8\pi-\beta^2})}\right]^{2-\frac{\beta^2}{4\pi}}, \tag{2.8}$$

valid in both the attractive and repulsive regimes and assumes the conformal field theory (CFT) normalisation for the cosine operator. In the attractive regime, the masses of the breathers $m_n$ can be expressed in terms of the soliton mass $M$ as

$$m_n = 2M\sin\frac{\pi\xi n}{2}, \tag{2.9}$$

with $n = 1,...,\lfloor 1/\xi \rfloor$. It is important to mention that the mass gap is given by the mass of the 1st breather $m_1$ if $\xi \le \frac{1}{2}$ and by the mass of the soliton $M$ if $\xi > \frac{1}{2}$.

As well known, this theory is integrable and hence admits factorised scattering. The corresponding $S$-matrices are known exactly [80], but their explicit form is not necessary for our present purposes. Nevertheless, it is important to say a few words about the classical counterpart of the model, which is an integrable classical field theory. The classical theory can be defined by the same action (2.1) and the coupling $\lambda_{\text{cl}}$ (which we distinguish from the coupling constant of the quantum theory) is conventionally chosen as

$$\lambda_{\text{cl}} = \frac{m^2}{\beta^2}, \tag{2.10}$$

where $m$ is a mass scale. Based on the finite energy, static and time-dependent, solutions of the corresponding equation of motion (EOM)

$$\partial_t^2\varphi(x,t) - \partial_x^2\varphi(x,t) + \frac{m^2}{\beta}\sin[\beta\varphi(x,t)] = 0, \tag{2.11}$$

one can talk about configurations including soliton and/or anti-soliton excitations, as well as breathers, irrespective of the magnitude of $\beta$. The solitons and anti-solitons interpolate between neighbouring vacua of the cosine potential, $\varphi(\infty,t) - \varphi(-\infty,t) = \pm 2\pi/\beta$, respectively, so their topological charge is $+1$ and $-1$. The neutral breather is a time-periodic configuration that can be viewed as a bound state of a soliton and an antisoliton. Unlike the quantum case, the breather mass is not quantised in the classical theory. Nevertheless, one can make an important connection between the mass scale $m$ of the classical theory and the first breather mass $m_1$ of the quantum theory, at least in the small $\beta$ regime. For both the classical and quantum theories, in the limit $\beta \to 0$ the Klein–Gordon theory is recovered. On the classical side, the boson mass is simply $m$, whereas on the quantum side the elementary boson is identified with the first breather. We can therefore identify the two mass scales $m$ and $m_1$ when comparing the quantum and classical results.

Finally, there is one last aspect worth emphasising in preparation for the following investigations. An important quantity is the zero-mode of the field that can be formally defined as

$$\lim_{L\to\infty}\frac{1}{L}\int_{-L/2}^{L/2}\mathrm{d}x\,\varphi(x,t) = \varphi_0, \tag{2.12}$$

both in the quantum and in the classical theory. In the quantum theory, however, the zero mode, as well as the canonical field $\hat{\varphi}$, are only well-defined when their exponentials are considered. In particular, only operators

$$e^{ik\beta\hat{\varphi}_0}, \qquad e^{ik\beta\hat{\varphi}(x,t)}, \tag{2.13}$$

with some integer $k$ are well defined since the fundamental field in the QFT is compactified as

$$\hat{\varphi} \equiv \hat{\varphi} + \frac{2\pi}{\beta} k. \tag{2.14}$$

In contrast, in the classical theory both the zero mode and the classical field $\varphi$ are well-defined quantities themselves. In many cases, nevertheless, it is instructive to consider the elementary field $\hat{\varphi}$ also at the quantum level, provided its meaning is appropriately specified. In the following, we consider the expectation value of $\hat{\varphi}$ defined as

$$\langle \hat{\varphi}(x, t) \rangle := \int_{-\infty}^{x} dx' \langle \partial_{x'} \hat{\varphi}(x', t) \rangle. \tag{2.15}$$

This quantity is especially useful for making comparisons with the classical field $\varphi(x, t)$ but lacks information about the zero mode $\hat{\varphi}_0$.

## 2.2 Inhomogeneous initial states and time evolution

In the following we discuss a natural protocol to obtain inhomogeneous initial states. Conforming to the quantum quench paradigm, we consider the initial state as the ground state of the Hamiltonian

$$\hat{H}_{\text{inhom}} = \hat{H}_{\text{sG}} - \int dx\, \partial_x \hat{\varphi}(x, t)\, j'(x), \tag{2.16}$$

where $j(x)$ is a static source term or external field, $j'$ denotes its spatial derivative. At first glance, the use of a derivative field $j'(x)$ in (2.16) might seem an unnecessary complication in our conventions. Nevertheless, as shortly demonstrated, this choice comes very handy and natural for our later investigations, when the classical theory is also considered. As clear from (2.16), an inhomogeneous external field is coupled to the spatial derivative of the fundamental field in the Hamiltonian above. According to (2.2), this field is equivalent to coupling the inhomogeneous external source $j'(x)$ to the topological charge density, therefore $j'(x)$ can also be regarded as an external chemical potential. Since our numerical methods allow us to treat this problem only in finite volume, it is useful to write our inhomogeneous sine–Gordon Hamiltonian as

$$\begin{aligned}
\hat{H}_{\text{inhom}} &= \hat{H}_{\text{sG}} + \hat{H}_j \\
&= \int_{-L/2}^{L/2} dx \left\{ \frac{1}{2} \hat{\Pi}(x, t)^2 + \frac{1}{2} (\partial_x \hat{\varphi}(x, t))^2 - \lambda \cos(\beta \hat{\varphi}(x, t)) \right\} \\
&\quad - \frac{2\pi}{\beta} \int_{-L/2}^{L/2} dx\, \hat{\rho}(x, t)\, j'(x),
\end{aligned} \tag{2.17}$$

where the conjugate momentum $\hat{\Pi}(x, t)$ equals $\partial_t \hat{\varphi}(x, t)$, and periodic boundary conditions are imposed. For our investigations, we choose the ground state $|0\rangle_j$ of the above Hamiltonian (2.17) as the initial state of the time evolution.

We shall also consider the classical analogue of the quantum problem, where the classical Hamiltonian $H_{\text{inhom}}[\varphi]$ (energy functional) equals (2.17) with $\lambda \to \lambda_{\text{cl}}$ and with the quantum fields replaced by classical ones. The 'classical ground state' $\varphi_j(x)$ is then the lowest energy configuration of the classical Hamiltonian

$$H_{\text{inhom}}[\varphi_j] = \text{minimum}. \tag{2.18}$$

The lowest energy configuration has zero momentum and can be easily obtained using the variation principle. It is a solution of the boundary value problem

$$\varphi_j''(x) = \frac{m^2}{\beta} \sin[\beta \varphi_j(x)] + j''(x), \qquad (2.19)$$

with the $x \in [-L/2, L/2]$, and the periodic boundary conditions

$$\begin{aligned} \varphi_{\text{odd}}(-L/2) &= \varphi_{\text{odd}}(L/2) = 0, \\ \varphi_{\text{even}}'(-L/2) &= \varphi_{\text{even}}'(L/2) = 0, \end{aligned} \qquad (2.20)$$

as long as the external field $j(x)$ is a parity odd function, which is true in our case as discussed soon. In (2.20) we use the usual parity decomposition

$$f_{\text{even/odd}}(x) = \frac{1}{2} \left( f(x) \pm f(-x) \right). \qquad (2.21)$$

Introducing a rescaled field variable $\tilde{\varphi} = \beta \varphi$ and rewriting (2.19) as

$$\tilde{\varphi}_j''(x) = m^2 \sin[\tilde{\varphi}_j(x)] + A\beta j_0''(x), \qquad (2.22)$$

with $j_0$ the external field of unit amplitude, we see that the classical boundary value problem is controlled by the parameter $A\beta$, where $A$ parameterises the magnitude of the external field $j$. In our subsequent investigation we change both the interaction parameter $\beta$ and the amplitude of the external field; in the classical theory, nevertheless, these two choices are completely equivalent. Finally, it is important that both quantum and classical Hamiltonians (2.17) are bounded from below, as expected, which can be easily seen by replacing $\hat{\varphi} \rightarrow \hat{\varphi} + j(x)$.

In our quench protocol, the subsequent time evolution is governed by the homogeneous sine–Gordon Hamiltonian defined in Eq. (2.17) as $e^{-it\hat{H}_{\text{sG}}}|0\rangle_j$ and the expectation value of $\hat{\rho}(x,t)$ can be expressed as

$$\langle \hat{\rho}(x,t) \rangle_j = {}_j\langle 0|e^{it\hat{H}_{\text{sG}}} \hat{\rho}(x) e^{-it\hat{H}_{\text{sG}}} |0\rangle_j, \qquad (2.23)$$

where $\langle \hat{\rho}(x,t) \rangle_j$ is a shorthand to indicate the dependence of the initial state on $j(x)$. Multiplying this expectation value by $2\pi/\beta$ and integrating over $x$ we obtain $\langle \hat{\varphi}(x,t) \rangle_j$ as defined in (2.15).

To obtain the classical counterpart of the problem, the classical EOM (2.11) is integrated with the initial conditions $\varphi(x,0) = \varphi_j(x)$ and $\partial_t \varphi(x,0) = 0$, from which $\varphi_j(x,t)$ or $\partial_x \varphi_j(x,t) \propto \rho_j(x,t)$ are naturally obtained and the subscript $j$ stresses the starting initial conditions. Finally, we note that in our numerical simulations the dimensionless quantities are obtained by setting Planck's constant and the speed of light to 1 and measuring everything in appropriate powers of the first breather mass $m_1$ in the quantum case, and of the mass $m$ in the classical setting.

## 3  Methods

Our numerical method to compute the inhomogeneous initial state $|0\rangle_j$ and its time evolution $e^{-it\hat{H}_{\text{sG}}}|0\rangle_j$ is based on a very efficient realisation of the Truncated Conformal Space Approach (TCSA). TCSA is a numerical method to study perturbed conformal field theories (PCFTs), especially in 1+1 D, originally introduced in [69] and its essence can be summarised in a relatively simple way. As well known, PCFT is a paradigmatic approach to massive quantum field theories, regarding them as perturbations of their ultra-violet (UV) fixed point conformal

field theories [81] by appropriate relevant operators. In this terminology, perturbation does not necessarily mean that the coupling is weak, and in fact – especially in models with one space dimension – this paradigm is powerful enough to enable non-perturbative studies at strong coupling as well.

In TCSA the theory of interest is considered in a finite volume $L$, which results in a discrete spectrum of the unperturbed CFT. To obtain a finite dimensional Hilbert space, the spectrum is truncated to a finite subspace by introducing an upper energy cut-off parameter $e_c$. Crucially, in CFTs it is often possible to calculate exact finite volume matrix elements of the perturbing fields and various operators of interest in the truncated Hilbert space. In the end, therefore, computing the spectrum and other physical quantities in many cases reduces to relatively simple manipulations with finite dimensional matrices. In addition, the inevitable cut-off dependence of various physical observables can be brought under control and also (approximately) eliminated by renormalisation group methods [82–86]. In particular, the extrapolation procedure is well-established for the expectation values of the derivative field $\partial_x \hat{\varphi} \propto \hat{\rho}$ in the ground state and excited states [87], and as we argue in Appendix A.3, the standard extrapolation procedure applies for inhomogeneous states as well under reasonable circumstances. Nevertheless, the naive extrapolation is generally not expected to be applicable for time evolved quantities, although it is known to work in some limited cases [58, 72]. In this work, therefore, we primarily use extrapolation for the study of the initial state, and for time evolving quantities we instead present the data corresponding to the highest possible cut-off. However, the extrapolated expectation values are displayed in Appendix A.5 for completeness.

TCSA can be easily applied to the sine–Gordon theory which can be regarded as the relevant perturbation of the free massless, compactified bosonic CFT by the vertex operator $\cos(\beta\varphi)$. For more details, we refer the reader to Appendix A.

# 4 Results

Our main goal in this work is to study the time evolution of the topological charge density dictated by the unitary time evolution of the homogeneous sine–Gordon model $\hat{H}_{\text{sG}}$ when the initial state is the ground state of the inhomogeneous sine–Gordon theory $\hat{H}_{\text{inhom}}$. In the inhomogeneous Hamiltonian, the topological charge density is coupled to an external field $j'(x)$ according to (2.17), which is chosen to be localised so that the inhomogeneity is localised. As a representative choice for this situation, we use

$$j'(x) = \frac{A}{\sqrt{2\pi}\sigma m_1} \exp\left(-\frac{x^2}{2\sigma^2}\right) - \frac{A}{\ell}\operatorname{erf}\left(\frac{\ell}{2\sqrt{2}\sigma m_1}\right), \qquad (4.1)$$

that is, the derivative field is a Gaussian centred at the middle of the interval $[-L/2, L/2]$. Here $\ell$ denotes the dimensionless length of the system $\ell = m_1 L$, and $m_1^{-1}A$ and $m_1\sigma$ are dimensionless parameters controlling the amplitude and the width of the Gaussian bump. A similar source term was considered in [88] where it was reported to result in 'super-soliton' behaviour[1] and initial Gaussian temperature profiles were used in other integrable models too [89, 90]. Throughout this work, we impose the following neutrality condition for the external field

$$\int_{-L/2}^{L/2} \mathrm{d}x\, j'(x) = 0, \qquad (4.2)$$

which is achieved by subtracting the constant term $A/\ell \operatorname{erf}\left(\ell/(2\sqrt{2}m_1\sigma)\right)$ from the Gaussian bump.

---

[1] However, we note that TCSA with periodic boundary conditions used in this work is not able to reproduce the quench protocol of [88] which requires quenching the $\beta$ parameter or equivalently, the compactification radius $R$.

Table 4.1: The zero mode in the quantum inhomogeneous state from TCSA with $e_c = 30$, and in the classical lowest energy configuration (last column) when $m_1 L = 20$, $m = m_1$. $\langle \mathcal{O} \rangle_j$ denotes the expectation value of operator $\mathcal{O}$ in the inhomogeneous initial state, while $\sigma\left[\langle \mathcal{O} \rangle_j\right]$ is the standard deviation characterising the fluctuations.

| $A\beta_{FF}/m_1 = 18$ $m_1 \sigma = 2/3$ | $\langle \cos \beta \hat{\varphi}_0 \rangle_j$ | $\langle \cos^2 \beta \hat{\varphi}_0 \rangle_j$ | $\sigma\left[\langle \cos \beta \hat{\varphi}_0 \rangle_j\right]$ | $\langle \sin \beta \hat{\varphi}_0 \rangle_j$ | $\langle \sin^2 \beta \hat{\varphi}_0 \rangle_j$ | $\sigma\left[\langle \sin \beta \hat{\varphi}_0 \rangle_j\right]$ | $\beta \varphi_0$ (classical) |
|---|---|---|---|---|---|---|---|
| $\beta = \beta_{FF}/2.3$ | 0.9666 | 0.9366 | 0.0471 | 0 | 0.0633 | 0.2515 | 0 |
| $\beta = \beta_{FF}/2.0$ | -0.7223 | 0.6280 | 0.3259 | 0 | 0.3700 | 0.6083 | $\pi$ |
| $\beta = \beta_{FF}/1.7$ | -0.7030 | 0.6112 | 0.3420 | 0 | 0.3868 | 0.6219 | $\pi$ |
| $\beta = \beta_{FF}/\sqrt{5/2}$ | -0.6885 | 0.6011 | 0.3564 | 0 | 0.3971 | 0.6302 | $\pi$ |
| $\beta = \beta_{FF}/1.5$ | -0.6759 | 0.5930 | 0.3689 | 0 | 0.4053 | 0.6366 | $\pi$ |
| $\beta = \beta_{FF}/\sqrt{2}$ | -0.6589 | 0.5830 | 0.3858 | 0 | 0.4152 | 0.6444 | $\pi$ |
| $\beta = \beta_{FF}/1.1$ | -0.1997 | 0.5084 | 0.6845 | 0 | 0.4867 | 0.6976 | 0 |
| $\beta = \beta_{FF}$ | 0.0521 | 0.5004 | 0.7055 | 0 | 0.4922 | 0.7016 | 0 |

Our choice is further motivated by the fact that Eq. (4.2) implies that the ground state of the inhomogeneous problem lies in the $Q = 0$ (zero winding number or topological charge) sector, which is the most natural and relevant setting to investigate (see Appendix A for a more detailed explanation). Nevertheless, by our methods non-neutral sectors can be studied as well and satisfying the neutrality condition (4.2) could be achieved by various external field profiles such as two Gaussian bumps with opposite signs in their amplitudes. Although this latter choice is certainly interesting as well, it would make the study of time dependent quantities, such as the evolution of the bump in the topological charge density, more difficult. This is due to the limitations introduced by the finite volume: with a single Gaussian bump in $j'(x)$ instead of two, the time evolution can be followed for longer times before collisions by excitations propagating around the finite volume take place. The presence of a constant background value in $j'(x)$ in finite volume is also a perfectly sensible physical choice in its own right. Moreover, it leads to remarkable phenomena as shortly demonstrated.

## 4.1 Inhomogeneous initial states

Before studying the time evolution, it is natural to compare the initial expectation value $\langle \hat{\rho}(x) \rangle_j$ in the quantum inhomogeneous initial state to the classical profile $\rho_j(x)$ defined via Eqs. (2.18,2.19). We considered three pairs of values for $A$ and $\sigma$ chosen as $(A\beta_{FF}/m_1, m_1\sigma) = (12, 1/3), (15, 4/9)$ and $(18, 2/3)$, and various values of the coupling $\beta$. Note that the number of breather species is given by the integer part of $\xi^{-1}$ where $\xi$ is defined in Eq. (2.4). All values used here are taken from the attractive regime $\xi \leq 1$. Values of the coupling where $\xi^{-1}$ is integer correspond to reflectionless points where the otherwise non-diagonal scattering of the soliton/antisoliton excitations becomes diagonal. To make sure that our conclusions are independent of this feature, we chose couplings both at reflectionless and generic points. In addition, we fix the dimensionless length $m_1 L = \ell$ to 20. It is important to stress that this volume is understood in units of the first breather mass which only equals the mass gap for $\beta \leq \beta_{FF}/\sqrt{2}$. We also investigated couplings $\beta_{FF}/\sqrt{2} \leq \beta < \beta_{FF}$. In such cases, the volume $\ell = 20$, when re-expressed in terms of the mass gap $M$ as $ML$, decreases from 20 to 10 as $\beta$ goes from $\beta_{FF}/\sqrt{2}$ to $\beta_{FF}$.

Varying $\beta$ with $\ell$, $A$ and $\sigma$ fixed, an obvious transition can be seen which is reflected in both the initial profiles and in the expectation values of the zero mode $\hat{\varphi}_0$. This is demonstrated in Fig. 4.1 and Table 4.1 for the case $A\beta_{FF}/m_1 = 18$ and $m_1\sigma = 2/3$.

The observed changes in the profiles and the zero mode (expectation) values are surprisingly sharp and abrupt. In fact, the transition is completely discontinuous in the classical

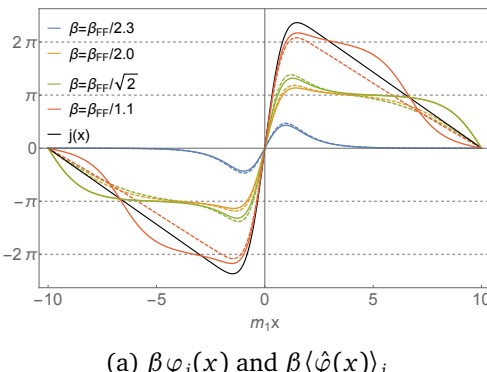

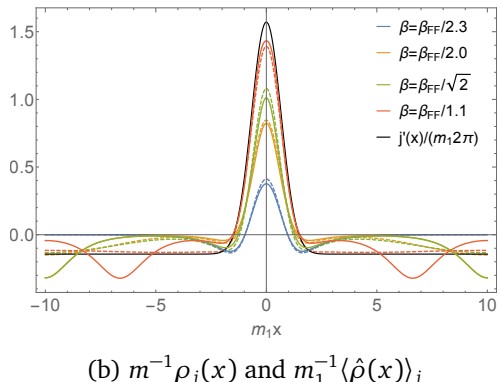

(a) $\beta\varphi_j(x)$ and $\beta\langle\hat{\varphi}(x)\rangle_j$

(b) $m^{-1}\rho_j(x)$ and $m_1^{-1}\langle\hat{\rho}(x)\rangle_j$

Figure 4.1: Comparing quantum and classical profiles for different values of the coupling $\beta$.
(a) The QFT expectation values $\langle\beta\hat{\varphi}(x)\rangle_j$ (dashed lines) and the classical lowest energy configurations $\beta\varphi_j(x)$ (continuous lines).
(b) The corresponding topological charge densities $m_1^{-1}\langle\hat{\rho}(x)\rangle_j$ in the quantum case (dashed lines) and $m^{-1}\rho_j(x)$ in the classical case (continuous lines).
The parameters are $\ell = m_1 L = 20$, $m = m_1$, $\beta_{\text{FF}}A/m_1 = 18$, $m_1\sigma = 2/3$; different $\beta$ values are shown with different colours and $\beta_{\text{FF}} = \sqrt{4\pi}$. The classical solutions for the two intermediate $\beta$ values have $\beta\varphi(0) = \pi$ and are shifted by $-\pi$. The TCSA profiles $\langle\hat{\rho}(x)\rangle_j$ were extrapolated using cut-offs $e_c = 24, 26, 28$ and 30, and the corresponding profiles $\beta\langle\hat{\varphi}(x)\rangle_j$ were obtained by spatial integration of $2\pi\langle\hat{\rho}(x)\rangle_j = \beta\langle\partial_x\hat{\varphi}(x)\rangle_j$, fixing the zero mode by requiring the result to vanish at the origin $x = 0$.

system even though the volume is finite as demonstrated by Fig. 4.1 and the last column of Tab. 4.1 displaying the classical value of the zero mode $\beta\varphi_0$. In the classical theory, depending on the interaction strength, the value of the zero mode changes abruptly from 0 to $\pi/\beta$ and back and the initial profile changes accordingly. With some differences in the details, this behaviour is also closely mirrored in the quantum theory, as can be seen by comparing the values of $\langle\cos\beta\hat{\varphi}_0\rangle_j$ and $\langle\sin\beta\hat{\varphi}_0\rangle_j$ in the 2nd and 5th column of Table 4.1 with the classical values for $\beta\varphi_0$ listed in the last column. The parallels between the behaviour of the classical and quantum initial state are also apparent from Fig. 4.1. The difference between the quantum and the classical initial state increases with $\beta$, as expected. It is also interesting to notice that the quantum initial state is smoother (has less spatial variation) than the classical, which can be easily understood since sharp localised features in classical quantities are expected to be washed out by quantum fluctuations.

The transition in the initial state can easily be understood by classical considerations, which we briefly review. When the cosine potential is neglected, the fundamental field $\varphi(x)$ follows the external source $j(x)$. Switching on the interaction, this tendency of the field is of course hindered by the energy cost due to the potential. One can examine this effect by either fixing $\beta$ and varying the amplitude $A$ of the external field or the other way around (in accordance with the rescaled classical equation of motion for the inhomogeneous problem (2.22)), with the result that above a given value of $\beta A$ it becomes energetically more favourable to shift the profiles $\varphi_j(x)$ by $\pi/\beta$, which guarantees that the field values can sit in two adjacent vacua of the cosine potential over extended spatial regions. In Fig. 4.1 and in Tab. 4.1 this transition happens between $\beta_{\text{FF}}/2.3$ and $\beta_{\text{FF}}/2.0$. The profiles on the two sides of this transition are shown in the upper row of Fig. 4.2. Note that, interestingly, after the transition the part of the profiles opposite to the external source agree with an antisoliton profile to a very good

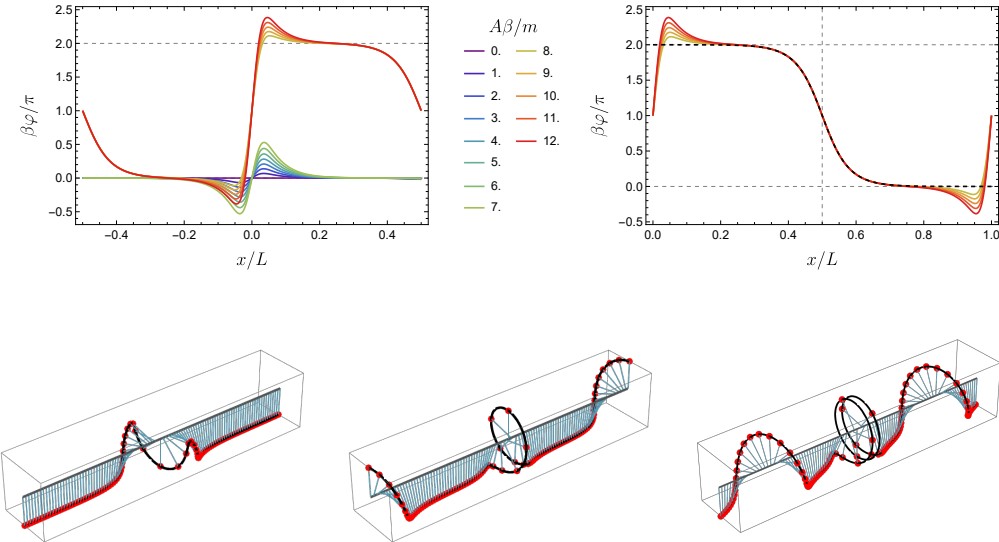

Figure 4.2: *Upper row left:* Classical initial states on the two sides of the first transition at $\ell = 20, m\sigma = 4/9$. Different curves correspond to different amplitudes as indicated in the legend. *Upper row right:* The profiles above the transition point ($A > A_{\mathrm{cr}}$ where $A_{\mathrm{cr}} \approx 7.856\, m/\beta$ (c.f. Table B.1) match perfectly around $x = L/2$ to the profile of an antisoliton in a rotated frame. *Lower row:* Illustration of the realisation of three types of classical initial states on a chain of coupled pendulums obeying the discretised sine–Gordon equation. The leftmost one corresponds to the type of profiles observed for $A < A_{\mathrm{cr}}$, while the middle one corresponds to the type of profiles observed for $A > A_{\mathrm{cr}}$ which exhibit a $2\pi$-twist of the phase at the location of the external source and an antisoliton configuration located the antipodal point on the circle. If $A$ is increased even further, then above a second critical value $A_{\mathrm{cr2}} \approx 14.412\, m/\beta$ a second twist emerges in the middle together with a corresponding second antisoliton, as shown in the rightmost figure.

precision (see inset).

Further increasing the amplitude $A$ or $\beta$ even larger zero mode values $k\pi/\beta$ are expected, which are equivalent to $\beta\varphi$ being either zero or $\pi$ by periodicity. The data in Fig. 4.1, and in Tab. 4.1 show that the second transition happens between $\beta_{\mathrm{FF}}/\sqrt{2}$ and $\beta_{\mathrm{FF}}/1.1$. The above intuitive picture can be visualised by displaying the response of a classical pendulum chain (a discretised version of the sine–Gordon field theory) to an external source term as shown in the lower panel of Fig. 4.2. Studying the classical energy functional after solving Eq. (2.19) with a fixed $\beta\varphi_0 = 0, \pi$, one can easily confirm that the change is discontinuous in the classical theory and the exact transition values $A\beta$ can be determined numerically. For the specific case of the parameters $A\beta_{\mathrm{FF}}/m = 18, m\sigma = 2/3$, that is when the amplitude of the external field is fixed, the critical $\beta$ values are $\beta_{\mathrm{FF}}/2.045 \approx 1.734$ and $\beta_{\mathrm{FF}}/1.161 \approx 3.053$ as indicated in Tab. 4.2.

Considering the QFT expectation values of the zero mode and the field profiles, the above transition can be observed in the quantum case as well. Furthermore, the transition in the quantum theory shares many features of the classical model such as the similarity of the transition values themselves and also the shape of the profiles for small amplitude or $\beta$. This latter feature is easy to understand as for small perturbations the response to the external field is captured by the quantum and classical Klein–Gordon theory, giving the same result for field expectation values.

Table 4.2: Energies of the two configurations $\varphi_j(x)$ with $\beta\varphi_0 = 0$ and $\pi$ when $m\sigma = 2/3$ and $\ell = mL = 20$.

| $\beta A/m$ | 4 | 6 | 8 | 8.8 | 10 | 12 | 14 | 15.502 | 16 | 17 |
|---|---|---|---|---|---|---|---|---|---|---|
| $H_{\text{i.h.}}[\varphi_j(x)]/L,\ \varphi_0 = 0$ | -1.064 | -1.145 | -1.262 | -1.320 | -1.422 | -1.615 | -2.176 | -2.623 | -2.776 | -3.089 |
| $H_{\text{i.h.}}[\varphi_j(x)]/L,\ \varphi_0 = \pi/\beta$ | 0.749 | -0.888 | -1.190 | -1.320 | -1.524 | -1.892 | -2.295 | -2.623 | -2.737 | -2.974 |

Nevertheless, clear and important differences are also present when the quantum and classical cases are compared. First of all, it is important to stress that we have no evidence for a discontinuous transition in the quantum theory. However, we cannot unambiguously clarify this issue with our current numerical method, therefore we leave this question open at the same time emphasising that the quantum transition is remarkably steep.

Focusing on the expectation values of $\langle \cos\beta\hat{\varphi}_0 \rangle_j$ a large jump from 0.967 to $-0.722$ and then a somewhat smaller from $-0.659$ to $-0.2$ is observed when the classical zero mode jumps from 0 to $\pi/\beta$ and back. The fluctuations of the zero mode increase after each transition as illustrated by the data in Tab. 4.1. More pronounced differences can be observed in the field profiles. Whereas the classical and quantum profiles are similar for small amplitudes (or $\beta$'s), after both the first and the second transition values clear differences can be observed. The classical profiles $\varphi_j(x)$ tend to develop plateaux around $\pm k\pi/\beta$ to minimise the potential energy. In the quantum theory, the length of the plateaux in $\langle \hat{\varphi}(x) \rangle_j$ is shorter after the first transition and they are entirely absent after the second one. Furthermore, the spatial derivative of the field at $x = \pm L/2$ differs significantly from that of the classical field. The derivative approximately equals $j'(-L/2) = j'(L/2)$, which is not true in the classical model, at least in the range of parameters considered here.

For the particular value $j'(\pm L/2)$ of the derivative field, a possible explanation can be given based on the soliton content of the initial state and the correspondence between solitonic excitations and fermions in the quantum theory as demonstrated by the local density approximations in 4.2. The consecutive transitions can be naturally associated with the emergence of soliton-antisoliton pairs, which is reflected by the shape of both the classical and quantum profiles $\varphi_j(x)$ and $\langle \hat{\varphi}(x) \rangle_j$.

The equivalence between the sine–Gordon and the massive Thirring models reviewed in Subsec. 2.1 provides a natural link between solitons of the sine–Gordon and fermions of the Thirring model. At the coupling $\beta_{\text{FF}} = \sqrt{4\pi}$, the neutral sector of the sine–Gordon model can be mapped exactly to the neutral sector of free Dirac fermions, allowing us to obtain (numerically) exact results. In the fermionic picture, the external field couples to the Dirac charge density acting as a spatially varying chemical potential. The simplest approach is the local density approximation (LDA) which assumes that at each position the system is in local equilibrium set by the local chemical potential. Naturally, this approximation becomes more accurate for slower spatial variations. The LDA prediction for the density profile is (for a derivation c.f. Appendix C.5):

$$\langle \hat{\rho}(x) \rangle_j = \frac{1}{\pi} \begin{cases} \sqrt{(j'(x)/2 + \mu_0)^2 - M^2} & \frac{j'(x)}{2} + \mu_0 \geq M\,, \\ -\sqrt{(j'(x)/2 + \mu_0)^2 - M^2} & \frac{j'(x)}{2} + \mu_0 \leq -M\,, \\ 0 & \text{otherwise}\,, \end{cases} \tag{4.3}$$

where $\mu_0$ is set such that the integral $\langle \hat{\rho}(x) \rangle_j$ of is zero. Eq. (4.3) immediately implies that $\beta \partial_x \langle \hat{\varphi}(\pm L/2) \rangle = j'(\pm L/2)$ for large enough external fields. This finding is confirmed by the numerically exact computation of the (topological) charge density in the free Dirac fermion theory c.f. Fig. 4.3 (b).

In Fig. 4.3 the cut-off extrapolated TCSA profiles for couplings $\beta = \sqrt{4\pi}/1.7, \sqrt{2\pi}$ and $\beta = \sqrt{4\pi}$ (free fermion point) are compared to the results of the LDA and the numerically

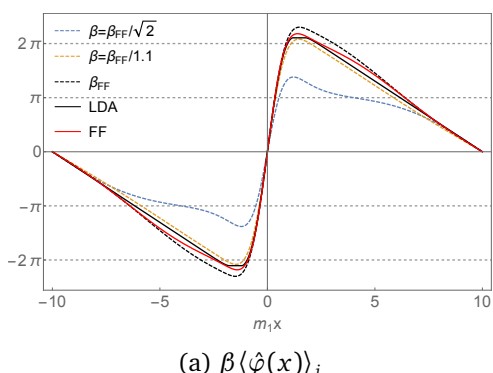

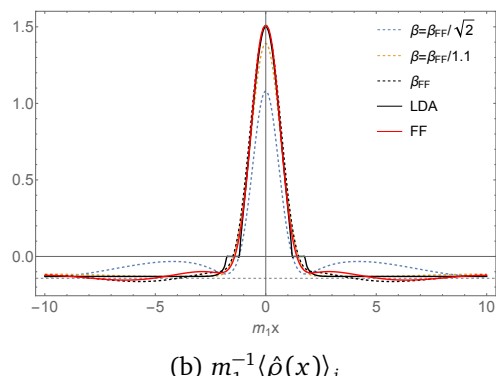

(a) $\beta \langle \hat{\varphi}(x) \rangle_j$

(b) $m_1^{-1} \langle \hat{\rho}(x) \rangle_j$

Figure 4.3: Comparing the quantum initial state to the exact free fermion and LDA results, with (a) showing the quantum expectation value of the field, while (b) displays the expectation value of the topological charge density together with the asymptotic value of $j'(\pm L/2)/(2\pi)$ with horizontal grey dashed line. The parameters are $\ell = m_1 L = 20, A\beta_{\text{FF}}/m_1 = 18, m_1\sigma = 2/3$. Blue, orange and black dashed curves are extrapolated TCSA data for $\beta = \beta_{\text{FF}}/\sqrt{2}, \beta_{\text{FF}}/1.1$ and $\beta = \beta_{\text{FF}}$ (free fermion point), while continuous black and red lines correspond to the LDA and the exact free fermion computation. For $\beta = \beta_{\text{FF}}$ the breather mass $m_1$ is taken as twice the soliton/fermion mass. TCSA profiles of $\langle \hat{\rho}(x) \rangle_j$ were extrapolated using cut-offs $e_c = 24, 26, 28$ and 30. In all cases, the corresponding profiles for $\beta \langle \hat{\varphi}(x) \rangle_j$ were obtained by spatial integration of $2\pi \langle \hat{\rho}(x) \rangle_j = \beta \langle \partial_x \hat{\varphi}(x) \rangle_j$, fixing the zero mode by requiring the result to vanish at the origin $x = 0$.

exact free fermion (FF) results. The good agreement between the LDA and FF results and the extrapolated TCSA data for the free fermion point is also a strong confirmation for the validity of our numerical method and the extrapolation. The TCSA profiles away from the FF point are still qualitatively similar to the FF result, which indicates that the underlying mechanism for the observed effects is related to the fermionic nature of the solitonic excitations in the quantum theory, which is absent in the classical case.

The initial state transition can also be observed for other values of the amplitude and width $(A\beta_{\text{FF}}/m_1, m_1\sigma) = (12, 1/3), (15, 4/9)$, as shown in Appendix B.2. These other parameter values are particularly important for our later investigations, and the behaviour of the transition highlights important distinction between the classical and quantum cases. Namely, in the classical theory the transition only depends on the combination $A\beta$ parameter for a fixed bump width, which is not expected to hold in the quantum theory. One obvious reason is the upper bound $\beta^2 = 8\pi$ corresponding to the BKT transition, above which the cosine operator is irrelevant in the renormalisation group sense. Nevertheless, at least in the range of parameters considered here, the transition in the quantum theory can still be described in terms of fixing $A$ and decreasing $\beta$, or alternatively fixing $\beta$ and increasing $A$. This is explicitly demonstrated by Fig. 4.4, where the amplitude of the field is varied for a fixed interaction $\beta = \beta_{\text{FF}}/\sqrt{2}$ and bump width $m_1\sigma = 2.3$, with the data clearly showing that the same transition is observed at the quantum level. This freedom of exchanging the role of the parameters $\beta$ and $A$ can have significant implications for experiments with a weak interaction parameter (small $\beta$), despite the limitations for the validity of the above interchangeability which are addressed by this present work. Namely, we see that strong-coupling phenomena can also be reproduced with smaller couplings $\beta$ by considering larger amplitudes for the external source.

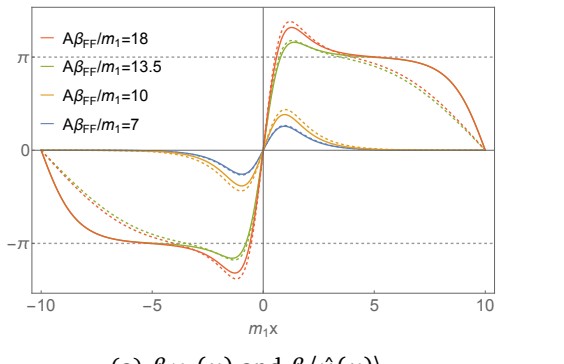

(a) $\beta\varphi_j(x)$ and $\beta\langle\hat{\varphi}(x)\rangle_j$

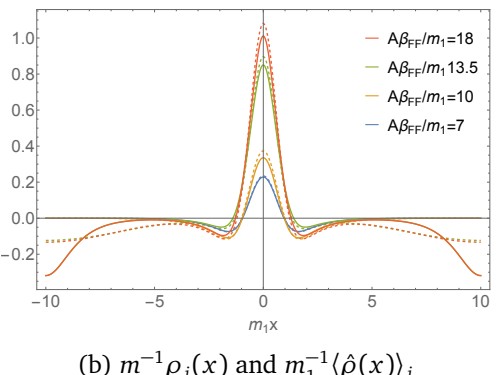

(b) $m^{-1}\rho_j(x)$ and $m_1^{-1}\langle\hat{\rho}(x)\rangle_j$

Figure 4.4: Comparing classical and quantum profiles for a fixed interaction $\beta$ and varying amplitudes $A$, with (a) showing the QFT expectation values $\langle\beta\hat{\varphi}(x)\rangle_j$ and the classical lowest energy configurations $\beta\varphi_j(x)$, while (b) displaying the topological charge densities $m_1^{-1}\langle\hat{\rho}(x)\rangle_j$ and $m^{-1}\rho_j(x)$. Continuous lines correspond to the classical case, while quantum results are shown with dashed lines. The parameters are $\ell = m_1 L = 20$, $m = m_1$, $\beta = \beta_{FF}/\sqrt{2}$, $m_1\sigma = 2/3$; different $A$ values are shown with different colours. The classical solutions for the two largest $A$ values have zero mode $\beta\varphi(0) = \pi$ and are shifted by $-\pi$. The TCSA profiles $\langle\hat{\rho}(x)\rangle_j$ were extrapolated using cut-offs $e_c = 24, 26, 28$ and $30$, and the profiles $\beta\langle\hat{\varphi}(x)\rangle_j$ were obtained by spatial integration of $2\pi\langle\hat{\rho}(x)\rangle_j = \beta\langle\partial_x\hat{\varphi}(x)\rangle_j$, requiring the result to vanish at the origin $x = 0$.

## 4.2 Limitations of the local density approximation, and the transition in the free massive Dirac theory

The local density approximation (LDA) is a standard approach that has been used for decades to describe physical systems with inhomogeneities. In particular, it has a fundamental role in the hydrodynamic description of one-dimensional integrable systems where integrability is violated by some spatial inhomogeneity [29, 91–94].

Therefore it is noteworthy to observe that LDA can break down for the sort of inhomogeneous states studied in this work. We demonstrate this fact by comparing the predictions of LDA and the exact FF computation for the (topological) charge density at the FF point. Fixing the width of the Gaussian bump in $j'$ and decreasing its amplitude, the exact computation reports a similar transition in the profile of $\langle\hat{\rho}\rangle_j$ to what was observed in the interacting regime of the sine–Gordon model. The LDA, nevertheless, is unable to reproduce the numerically exact profiles for external fields with small amplitude as demonstrated by Fig. 4.5, where the two dips next to the bump are absent in the LDA profiles, but a plateau emerges far away from the bump, which is not present in the exact profile. Limitations of LDA follow explicitly from Eq. (4.3) as well; for small enough $j$, no real chemical potential $\mu_0$ exists that could ensure the zero total charge condition. It is, nevertheless, important to stress again that the breakdown of LDA happens for small amplitudes of the external field, which corresponds to a rescaling only; the width of the bump is unchanged. These investigations are performed in the free massive Dirac theory in finite volume, thus without an interaction parameter. Based on our observations, nevertheless, one can infer the failure of LDA in the interacting regime of the sine–Gordon model as well.

The breakdown of the LDA can be easily understood from its nature which is essentially a mean field approach simplified further by neglecting gradient terms. The mean field approach is expected to fail for small densities where the quasi-classical approximation for the quantum field loses validity. In addition, neglecting the gradient terms limits the approach to slowly

varying source profiles. Whenever the source amplitude is too small or its spatial variation is too fast, LDA is not expected to reproduce the behaviour of the quantum field theory. In Appendix C.5, we present and carefully discuss the precise conditions for the applicability of the LDA formula (4.3). Note that the external field $j'(x)$ is regarded as a chemical potential $\mu(x)$ for the fermions. In terms of the external chemical potential, the conditions for the validity of the LDA can be formulated as

$$\left| \frac{\mu'(x)}{\mu(x)} \right| \ll M, \tag{4.4}$$

which expresses that the chemical potential changes slowly on microscopic quantum scales, and

$$\langle \hat{\rho}(x) \rangle \left| \frac{\mu(x)}{\mu'(x)} \right| \gg 1, \tag{4.5}$$

which guarantees the presence of a large enough number of particles in a 'fluid cell' of the system, and hence the validity of a mean field treatment. It is clear that $|\mu'(x)/\mu(x)|$ is insensitive to the variation of the amplitude $A$ of the chemical potential. However, $|\langle \hat{\rho}(x) \rangle \mu(x)/\mu'(x)|$ behaves for large $A$ as $A^2/A \propto A$ and hence for high enough amplitudes, the predictions of LDA are reliable as long as the condition (4.4), which is independent of the amplitude $A$, is satisfied. On the contrary, for small amplitudes $A$ of the chemical potential, the condition (4.5) is clearly violated.

Finally, we emphasise the presence of the transition to a Klein–Gordon-like initial state in the free Dirac theory as the magnitude of the external field is decreased. In fact, for small enough amplitudes of $j'$ and consequently for small particle density, the initial density profile is well described by the Klein–Gordon theory. This fact is demonstrated by panel (a) of Fig. 4.5, where for the lowest applied amplitude of the external field the density profiles $\langle \hat{\rho}(x) \rangle_j$ of the free fermion and Klein–Gordon theories (after appropriate rescaling) are compared. In the fermionic language of the theory, the initial profile can be understood as an effect of localised fermion and antifermion excitations where the localisation takes place for small densities. Nevertheless, an understanding of this phenomenon comes more naturally using the bosonic formulation of the Dirac theory via Eq. (2.17) with $\beta = \beta_{\mathrm{FF}}$. For small amplitude external fields the use of linear response theory is justified, and the expectation value of the charge density is expected to be linearly related to the external field and hence to be small. For a small amplitude external field, the quadratic term of the cosine operator in Eq. (2.17) dominates and the resulting effective theory is therefore equivalent to the Klein–Gordon model.

### 4.3 Post-quench time evolution

After the analysis of the initial profiles we turn to the study of the ensuing evolution under the homogeneous Hamiltonian $\hat{H}_{\mathrm{sG}}$. We simulate the dynamics using the TCSA method by implementing an expansion of the time evolution operator in terms of Chebyshev polynomials. The advantage of this approach is that it does not require the diagonalisation nor the computation of the exponential of the Hamiltonian but only its action on state vectors needs to be computed. We provide more details about the method in Appendix A.4. For comparison, we also solved the classical sine–Gordon partial differential equation (EOM) with initial condition given by the classical initial state.

Our results for the evolution of topological charge density are shown in the density plots of Fig. 4.6. In the upper figure the TCSA results are plotted while the lower one shows the classical dynamics. In these figures each column corresponds to a given value of $\beta$ given at the top and each row corresponds to a specific $(A, \sigma)$ pair.

In all cases, the dynamics is dominated by two ballistically propagating fronts originating from the initial inhomogeneity and travelling at the speed of light. However, it is immediately

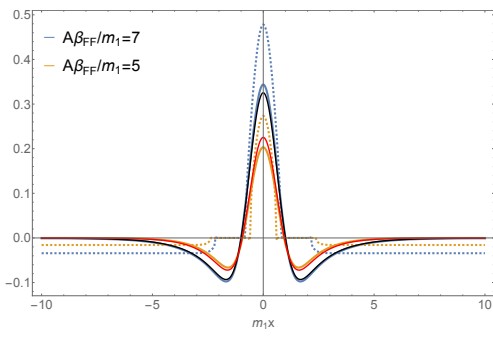

(a) $m_1^{-1}\langle\hat{\rho}(x)\rangle_j$ (small amplitudes)

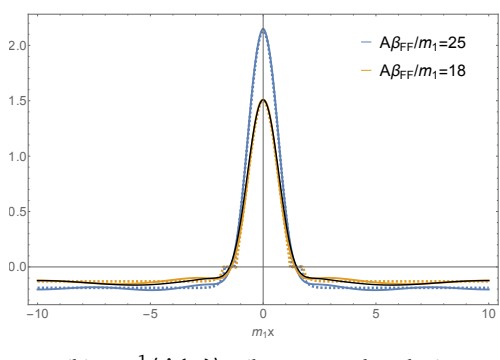

(b) $m_1^{-1}\langle\hat{\rho}(x)\rangle_j$ (large amplitudes)

Figure 4.5: The QFT expectation value of $\langle\hat{\rho}(x)\rangle_j$ at the free fermion point from LDA and the exact free fermion computation for four different amplitudes of the external field: $A\beta_{\text{FF}}/m_1 = 5$, $A\beta_{\text{FF}}/m_1 = 7$, and $A\beta_{\text{FF}}/m_1 = 18$, $A\beta_{\text{FF}}/m_1 = 25$. The bump-width in the external field is $m_1\sigma = 2/3$ in all cases. The coloured continuous lines correspond to the free fermion result, while the dashed ones to LDA. For $A\beta_{\text{FF}}/m_1 = 7$, and $A\beta_{\text{FF}}/m_1 = 18$ the TCSA profiles with continuous black lines are also shown. For the TCSA curves extrapolation was used based on the data with cut-offs $e_c = 24, 26, 28$ and $30$. The mass scale $m_1$ is understood as twice the fermion mass $M$. In subfigure (a) the rescaled Klein–Gordon initial profile $\langle\partial_x\varphi(x)\rangle_j/\sqrt{\pi}$ for $A\beta_{\text{FF}}/m_1 = 5$ is also displayed with a continuous red line.

apparent that there are two distinctly different kinds of behaviour. For small values of $\beta$ the behaviour is similar to the free boson case, and shows strong oscillations behind the front. For larger values of $\beta$ the space-time structure of the dynamics is similar to the free fermion behaviour and is much simpler: there are essentially two very stable counter-propagating bumps.

Comparing the quantum evolution shown in Fig. 4.6a to the classical one in Fig. 4.6b, it is interesting to observe that when the free-boson-like behaviour is realised in the quantum case, and the system is away from the initial state transition, the classical solution is very similar to the quantum dynamics. In contrast, the classical evolution is very different from the quantum evolution for those cases when the quantum theory exhibits free-fermion-like behaviour: classically some sort of revival takes place at the centre in addition to the ballistic front.

The evolution of the front profile in the co-moving frame is illustrated in Fig. 4.7, and its change with $\beta$ clearly follows the previously observed transition from bosonic to fermionic behaviour.

The smooth crossover between temporal behaviour characteristic for free bosonic vs. free fermionic degrees of freedom can be understood from the behaviour of the wave-function renormalisation factor [95]

$$Z = |\langle 0|\varphi(0)|p\rangle|^2 = (1+\xi)\frac{\pi\xi}{2\sin(\pi/2\xi)}\exp\left(-\frac{1}{\pi}\int_0^{\pi\xi} dx\,\frac{x}{\sin x}\right), \qquad (4.6)$$

displayed in Fig. 4.8 (a), where $|p\rangle$ denotes the one-particle state of the elementary boson particle (a.k.a. first breather). This shows that the spectral weight of the bosonic excitation is a monotonously decreasing function of the sine–Gordon parameter $\beta$, and exactly at the free fermion point $\beta^2 = 4\pi$ it goes to zero corresponding to the disappearance of all bosonic excitations from the spectrum which is indicated by their masses progressively crossing the two-soliton threshold as shown in Fig. 4.8 (b).

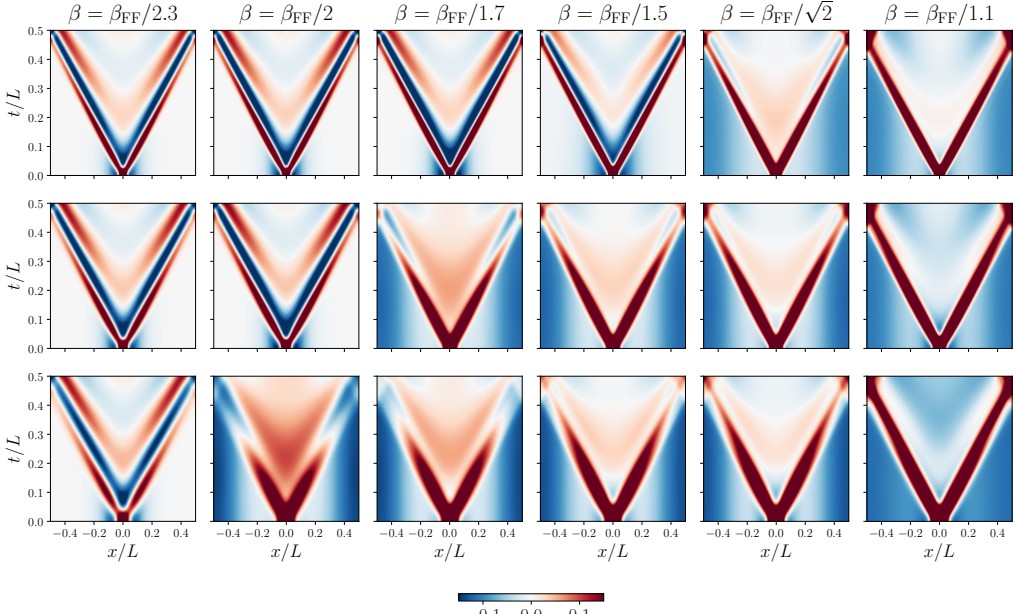

(a) TCSA simulation data for the quantum dynamics.

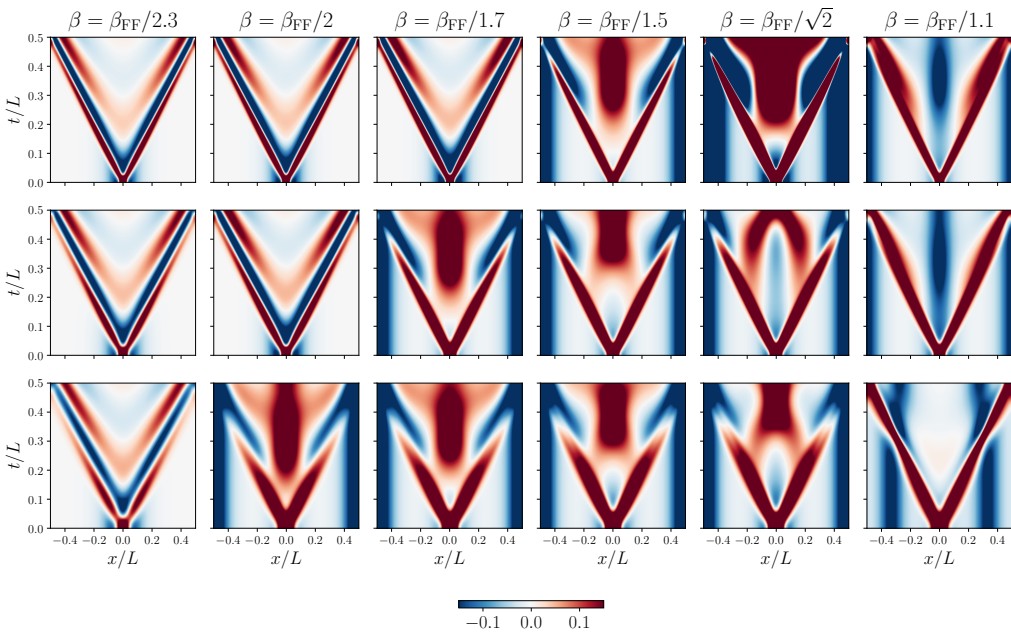

(b) Classical dynamics, starting from the classical initial profiles.

Figure 4.6: Space-time density plots of the time evolution of the topological density in the (a) quantum system (TCSA simulation) and in the (b) classical model. The plots correspond to each of the three different cases of external sources $j(x)$ and chosen $\beta$ values. (First row: $A\beta_{\mathrm{FF}}/m_1 = 12$, $m_1\sigma = 1/3$, second row: $A\beta_{\mathrm{FF}}/m_1 = 15$, $m_1\sigma = 4/9$, third row: $A\beta_{\mathrm{FF}}/m_1 = 18$, $m_1\sigma = 2/3$).

We also demonstrate that the features of the quantum dynamics discussed above such as the crossover between the two different types of behaviour for fixed external field and varying interaction strength, appear also when tuning the amplitude of the external field instead of the coupling $\beta$. Fixing the width for definiteness as $m_1\sigma = 2/3$, this is shown in Fig. 4.9, which also displays the exact free fermion dynamics and the corresponding TCSA simulation

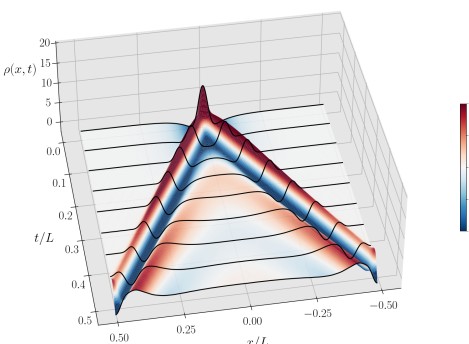

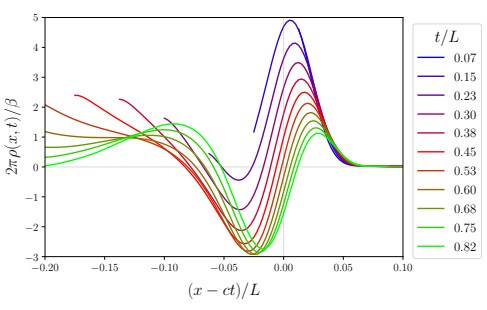

(a) 3D plot of the soliton density $\rho(x, t)$ corresponding to the case in the upper left corner of Fig. 4.6a ($\beta = \beta_{\text{FF}}/2.3$, $A\beta_{\text{FF}}/m_1 = 12$, $m_1\sigma = 1/3$).

(b) Plots of the right-moving front in the co-moving frame for the same parameters as in (a) at various times.

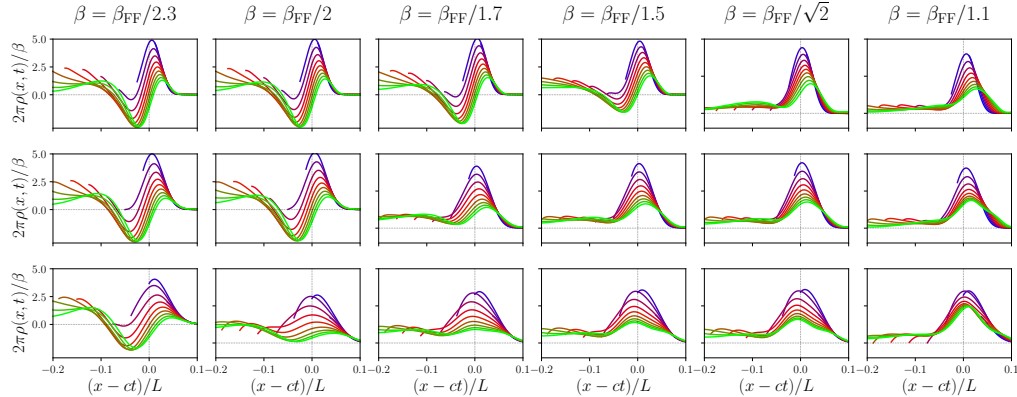

(c) Plots of the right-moving front at different times as in (b) for all the different cases in Fig. 4.6a (first row: $A\beta_{\text{FF}}/m_1 = 12$, $m_1\sigma = 1/3$, second row: $A\beta_{\text{FF}}/m_1 = 15$, $m_1\sigma = 4/9$, third row: $A\beta_{\text{FF}}/m_1 = 18$, $m_1\sigma = 2/3$).

Figure 4.7: Profile of the propagating front in the co-moving frame at different times, for the TCSA simulations shown in Fig. 4.6a. The qualitative shape of the front and especially its width are relatively stable with time for all parameters, but the height of the initial bump tends to decrease, while the depth of the accompanying dip which is present in the KG-like cases (once it is fully developed) remains practically invariant with time.

for comparison. The effect of changing the amplitude of the Gaussian bump for fixed interaction parameters was checked for another bump-width $m_1\sigma = 4/9$ resulting in a completely analogous behaviour to that of Fig. 4.9, which is not presented here for the sake of brevity.

It is noteworthy that, similarly to the behaviour of the initial state (cf. Fig. 4.5), the dynamics of the free massive Dirac fermion theory can be essentially Klein–Gordon-like for small enough external inhomogeneities. This can already be seen in Fig. 4.9, but for the sake of clarity we show the time evolutions in Fig. 4.10 for only the free bosonic/fermionic cases, rescaling the expectation value $\langle \partial_x \hat{\varphi}(x, t) \rangle_j$ by $1/\sqrt{\pi}$ in the Klein–Gordon case. The similarity in the time evolution can be easily understood by recourse to linear response theory and using the bosonic formulation of the Dirac theory Eq. (2.17), similarly to the argument in Sec. 4.2 concerning the case of the initial state.

The above observation has interesting implications for the fermionic degrees of freedom

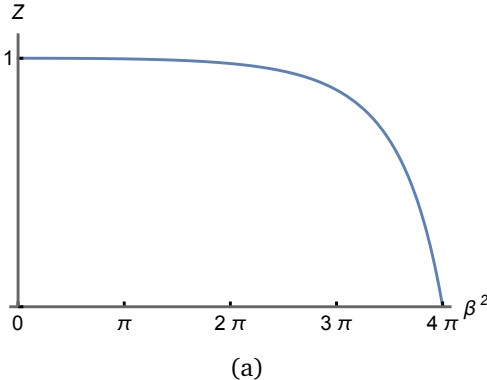
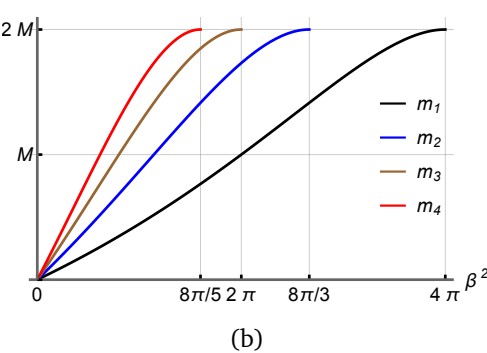

(a)

(b)

Figure 4.8: The wave-function renormalisation factor (a) and the masses of the first four breathers (in units of the soliton mass $M$) (b) as functions of the coupling $\beta$ in sine–Gordon theory.

and particularly for their time evolution. In the Klein–Gordon case, oscillations of the density profile behind the front are naturally attributed to the fundamental bosonic degrees of freedom in the theory. Indeed, it can be easily seen from Figs. 4.6 and 4.9 that the dominant oscillation frequency approximately equals $2\pi$ which is associated with a single boson at rest. In order to better demonstrate this fact, we explicitly plot $\langle \hat{\rho}(0,t) \rangle_j$ in Fig. 4.11 for the Klein–Gordon and free fermion cases with $A\beta_{\mathrm{FF}}/m_1 = 5$ and $m_1 \sigma = 2/3$. In the Dirac theory, the oscillation is instead associated with a fermion-antifermion pair which together make up a collective bosonic degree of freedom, and indeed the dominant frequency is given this time by twice the soliton mass (recall that at the free fermion point where the first breather is absent, the parameter $m_1$ was chosen as $2M$). Despite the absence of interactions this collective behaviour persists in the course of time evolution, at least up to intermediate times. However, it is also clear from Fig. 4.10 and Fig. 4.11 that the magnitude of the spatial oscillations in $\langle \hat{\rho}(x,t) \rangle_j$ decreases faster with time in the Dirac theory than in the Klein–Gordon model. This can be interpreted as indicating that the collective soliton-antisoliton (fermion/antifermion) pair is held together by the external source which determines the initial state, however in the homogeneous sine–Gordon dynamics at $t > 0$ they slowly drift away due to the absence of any binding force, while in the Klein–Gordon limit the bosonic particle is stable.

These findings are also important when interactions are present and the generic sine–Gordon model is considered, since they imply that the crossover in the dynamics (Fig. 4.9) for fixed interaction $\beta$ cannot be exclusively attributed to the inherent bosonic or fermionic degrees of freedom of the model for arbitrary interaction strengths. In particular, in the weakly attractive regime $\beta \lesssim \beta_{\mathrm{FF}}$, the $\beta$-dependent $Z$-factor (4.6) capturing the presence of fundamental bosonic degrees of freedom is small, and in the repulsive regime $Z$ is zero. Even so, the transition to a Klein–Gordon-like behaviour for low amplitude external fields is still expected to take place, in accordance with the observations at the free fermion point. In these cases the bosonic behaviour can be dominantly attributed to the collective behaviour of soliton and antisoliton excitations at low densities, similarly to the case of the free Dirac theory. In addition, it is also expected that the collective bosonic oscillations are suppressed as the time evolution progresses again similarly to the free fermion behaviour. As already mentioned, in the weakly interacting region close to the free fermion point $\beta \lesssim \beta_{\mathrm{FF}}$, even though the fundamental bosonic excitations are present in the spectrum, the matrix elements of physical quantities between such excited states are small. For this reason, for short and intermediate times their effect is negligible and the physics is dominated by the collective excitations. Since these excitations become suppressed as time elapses, at late times the presence of the breathers

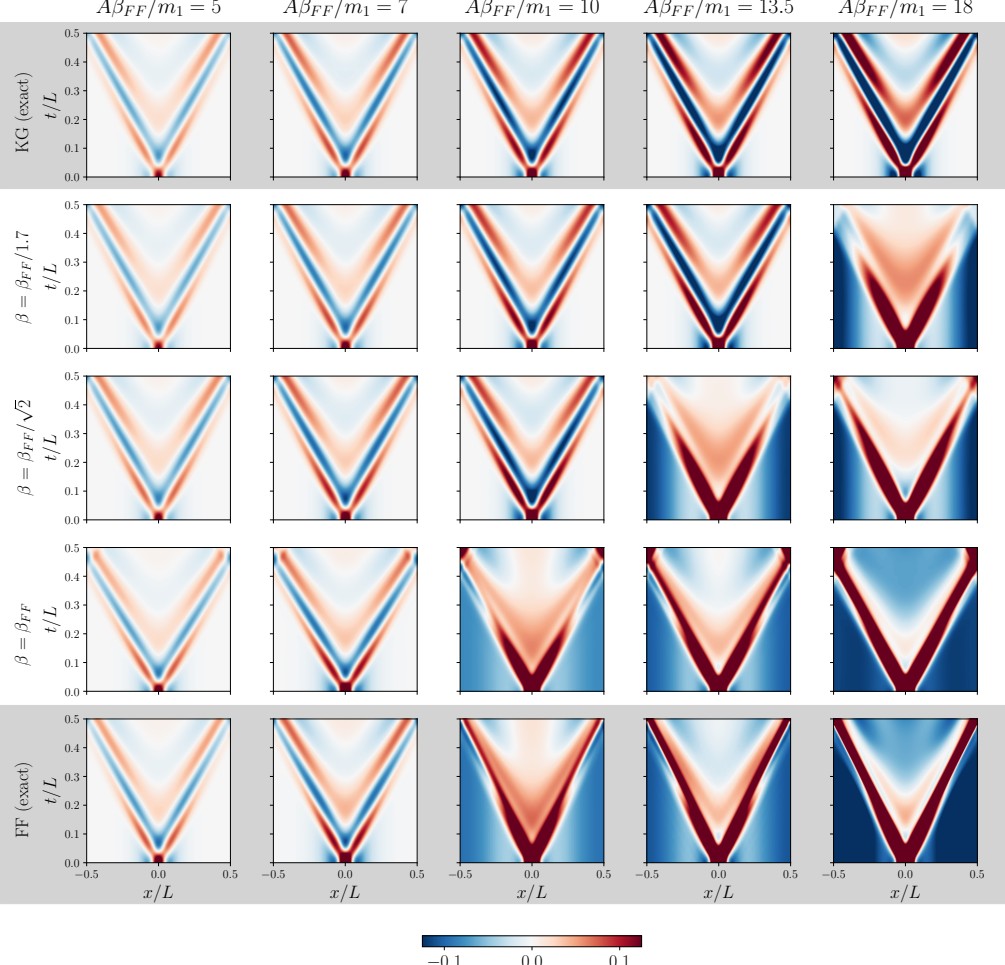

Figure 4.9: Time evolved QFT expectation value $m_1^{-1}\langle\hat{\rho}(x,t)\rangle_j$ for five different amplitudes of the external field with bump-width $m_1\sigma = 2/3$. The interaction parameters are $\beta = 0$ (Klein–Gordon theory), $\beta = \beta_{\text{FF}}/1.7$, $\beta = \beta_{\text{FF}}/\sqrt{2}$ and $\beta = \beta_{\text{FF}}$. For the computation of the free fermion dynamics, both TCSA and the numerically exact method was used. For the Klein–Gordon case $m_1^{-1}\langle\hat{\rho}(x,t)\rangle_j$ was obtained by rescaling $m_1^{-1}\langle\partial_x\hat{\varphi}(x,t)\rangle_j$ with $1/(1.7\sqrt{\pi})$ which corresponds to the $\beta = \beta_{\text{FF}}/1.7$ point shown in the row below. For the TCSA quantities, data with the cut-off $e_c = 30$ were used. (See the extrapolated version for TCSA in Appendix A.5.)

can be more important. Disentangling the effect of the different bosonic degrees of freedom close to the free fermion point could be accomplished by analysing the Fourier spectra corresponding to the dynamics of observables. Nevertheless, this is a non-trivial task because of the tiny mass difference between the soliton-antisoliton excitation ($2M$) and the first breather ($m_1 \approx 2M$). Moreover, such an analysis requires much larger times which are currently not accessible by our method and hence is not performed in this work.

However, deeper in the attractive regime of the model the fundamental bosonic degrees of freedom have a large weight as indicated by $Z$ in Fig. 4.8, and are expected to dominate the time evolution. This is shown by comparing the first two rows of Fig. 4.9, where the Klein–Gordon expectation value $\langle\partial_x\hat{\varphi}(x,t)\rangle_j$ is rescaled by $1/(1.7\sqrt{\pi})$ to match the soliton density of the sine–Gordon dynamics with $\beta = \beta_{\text{FF}}/1.7$. Note that the Klein–Gordon and sine–Gordon initial profiles match closely for the smallest amplitudes, and subsequent sine–Gordon time evolution also remains close to the Klein–Gordon counterpart, i.e. there is no

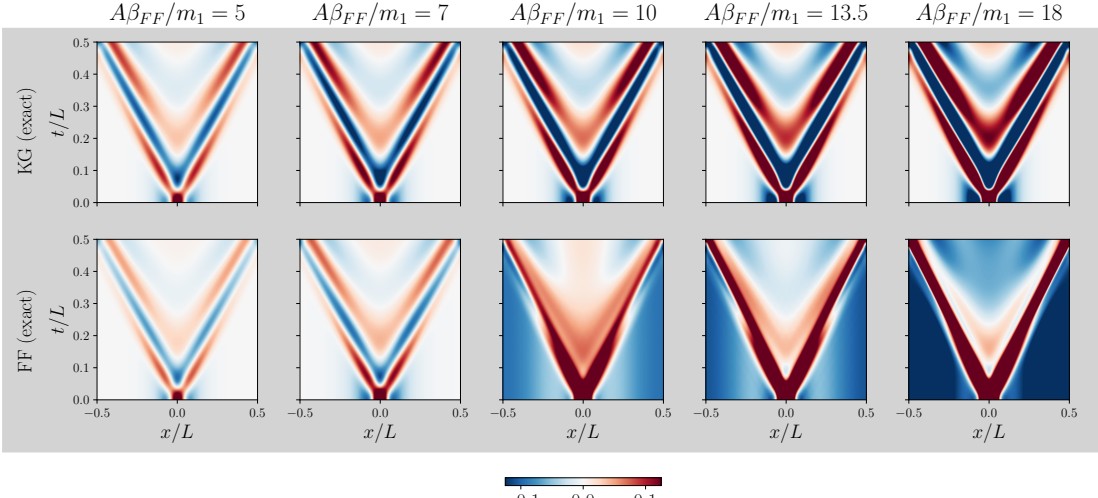

Figure 4.10: Comparison of the time evolution in the Klein–Gordon and free fermion limits for small amplitudes of the initial inhomogeneity.

additional temporal suppression compared to the Klein–Gordon dynamics, in contrast to what was observed at the free fermion point.

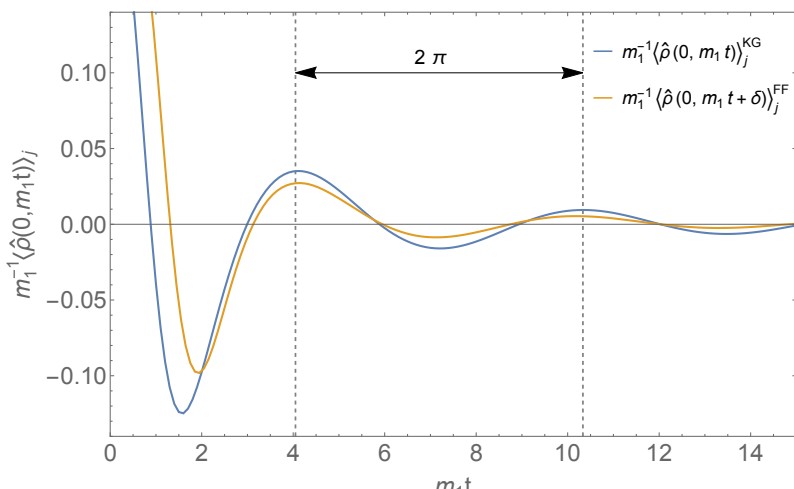

Figure 4.11: Klein–Gordon (blue line) and free fermion (orange line) time evolved expectation values of $m_1^{-1}\langle\hat{\rho}(0,t)\rangle_j$ at the fixed position $x = 0$ for an external field with amplitude $A\beta_{\mathrm{FF}}/m_1 = 5$ and bump-width $m_1\sigma = 2/3$. For the computation of the free fermion dynamics a numerically exact method was used and for the Klein–Gordon case $m_1^{-1}\langle\hat{\rho}(x,t)\rangle_j$ was obtained by rescaling $m_1^{-1}\langle\partial_x\hat{\varphi}(x,t)\rangle_j$ with $1/\sqrt{\pi}$ which corresponds to the $\beta = \beta_{\mathrm{FF}}$ point. For better transparency, the time argument in free fermion quantity $m_1^{-1}\langle\hat{\rho}(0,t)\rangle_j^{\mathrm{FF}}$ was shifted by $\delta = 0.45$. Note that the initial ratio $\langle\hat{\rho}(0,0)\rangle_j^{\mathrm{FF}}/\langle\hat{\rho}(0,0)\rangle_j^{\mathrm{KG}} = 0.901$.

We close this section with a discussion of the performance of TCSA based on comparisons with the exact free fermion dynamics. The reliability of the method was already benchmarked for the initial states, but estimating the quality of TCSA time evolution cannot rely merely on the performance of TCSA in equilibrium scenarios. Comparing the last two rows of Fig. 4.9 one can see that the free fermion dynamics is very nicely captured by TCSA. This finding is highly non-trivial, as sine–Gordon TCSA is known to be less convergent and precise as the

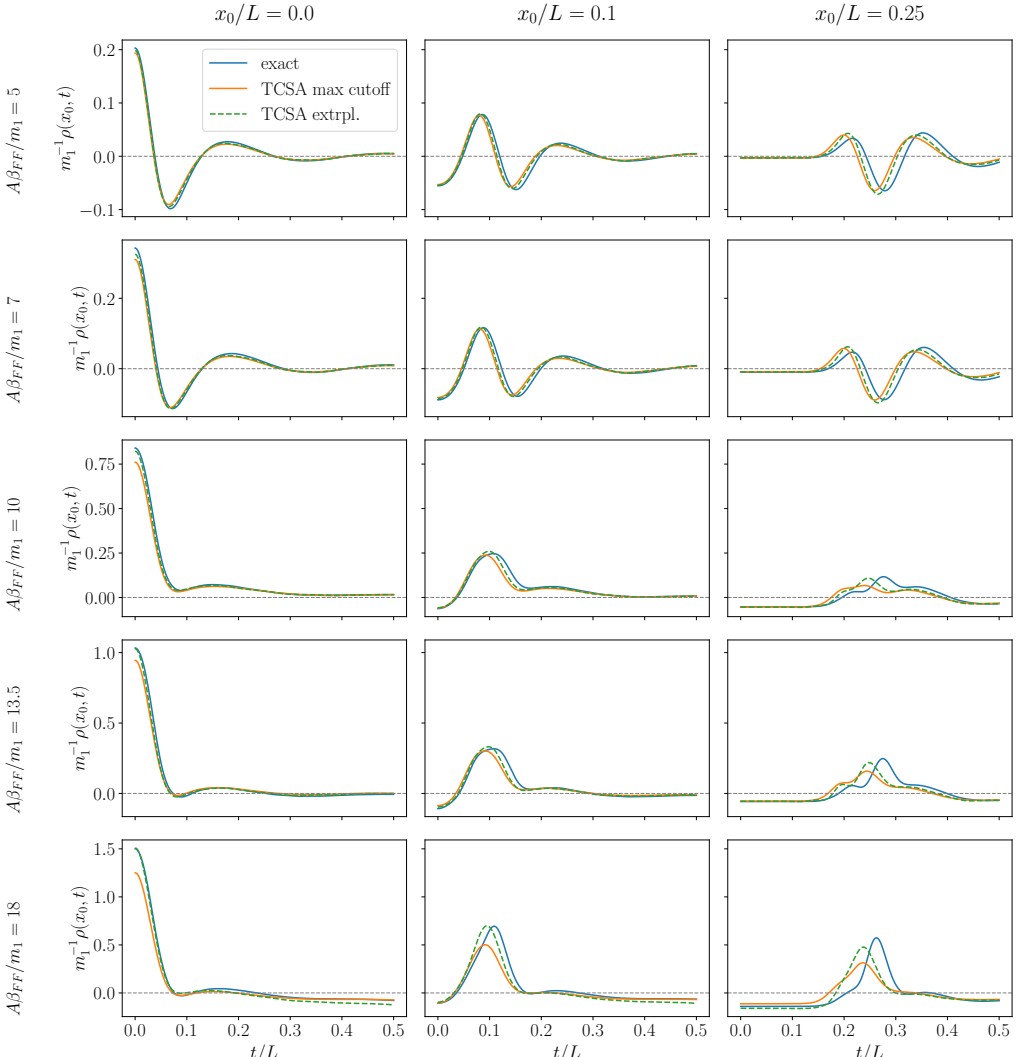

Figure 4.12: Comparison between TCSA and exact results for the dynamics at the free fermion point: The plots show the rescaled soliton density $m_1^{-1}\langle\hat{\rho}(x_0,t)\rangle_j$ as a function of time $t$ at three different fixed positions $x_0$ for five different values of the height $A$ of the initial bump. The different curves in each plot correspond to the exact free fermion time evolution (blue line) and the TCSA results at the maximum cut-off used in the simulations (red line) and after extrapolation (dashed green line) (see Appendix A.3 for details).

interaction parameter $\beta$ is increased, and in fact reaching the free fermion point is challenging [70,71]. One can, nevertheless, also observe a slight superluminal effect in the dynamics, which is enhanced by the increase in the magnitude of the external field and is attributed to the truncation that is known to introduce non-local effects [86]. As the interaction parameter $\beta$ is decreased, truncation effects are suppressed and this effect quickly becomes negligible as can be seen e.g. from the second and third rows of Fig. 4.9. We also present a quantitative comparison between the time evolution of the topological charge density at the free fermion point by plotting exact free fermion and TCSA results at some fixed positions $x/L$ as functions of time. In addition and for the sake of completeness, we also show the extrapolated TCSA curves. We stress again that the extrapolation procedure is not really justified for time evolving quantities, but it can be informative when estimating the accuracy of the TCSA. These compar-

isons are displayed in Fig. 4.12: based on the plots one can easily observe that TCSA captures well the qualitative features of the profiles as time elapses, although the amplitudes of the bumps and dips decrease with time compared to the exact free fermion results. This tendency of the TCSA data is partially mitigated by the extrapolation procedure. The already mentioned slight superluminality effect can be observed building up in the course of time evolution, and it is shown to affect both the raw and extrapolated TCSA data essentially to the same extent. These findings, more specifically, the observation that TCSA preserves the qualitative features of the exact free fermion dynamics, and the relatively slow quantitative deterioration of the TCSA results at the free fermion point together with the perfect quantitative match of TCSA simulations with Klein–Gordon dynamics for small $\beta$ (Fig. 1.1) strongly confirm the reliability of our TCSA simulations in the explored parameter regime and up to intermediate times.

## 5 Conclusions and outlook

In this work we considered the non-equilibrium dynamics of the sine–Gordon quantum field theory starting from an inhomogeneous initial state. The initial state was induced as the ground state in the presence of a localised external field with a Gaussian spatial profile coupled to the soliton charge density. Switching off the external field at time $t = 0$, the subsequent time evolution is governed by the homogeneous sine–Gordon Hamiltonian.

This setting is expected to be feasible in an experimental realisation of sine–Gordon theory, like the one based on coupled 1D quasi-condensates of ultracold atoms controlled by an atom chip. This experimental platform has been used to study quantum field dynamics leading among other results to the observation of the theoretically much anticipated Generalised Gibbs Ensemble [96]. The same platform can play the role of an analogue quantum simulator of the sine–Gordon model, as was demonstrated through the analysis of higher-order correlation functions in thermal equilibrium states [52, 97]. The sine–Gordon model provides an effective low-energy description of the relative phase field between the two quasi-condensates as their coupling due to hopping through the potential barrier separating them is of the form of an extended Josephson junction [50, 98]. The validity of the sine–Gordon description in out-of-equilibrium experiments is somewhat less clear and was challenged by the puzzling observation of a fast oscillation damping for initial states prepared with phase imprinting [55], an effect that was unexpected from the viewpoint of sine–Gordon theory [58]. However, it was recently shown that this theory-experiment deviation was due to the presence in early experiments of a parabolic confining trap [60, 61]. This is no longer a limitation in more recent experiments as uniform box-like traps have become feasible and have already been used for the observation of recurrences of quantum states [99, 100], an effect that relies strongly on the commensurability of the energy levels in Luttinger liquid (CFT) dynamics that is only possible in a homogeneous system. In fact it has been shown that with the use of digital micro-mirror devices (DMD) it is possible to implement an arbitrary space and time dependent external potential [101], providing the tunability that is necessary for the study of inhomogeneous systems and their dynamics.

On the other hand, it has been recently shown theoretically that solitons can be injected in the system using Raman coupling between the two condensates [102]. In this case the low-energy effective description of the system follows the Pokrovsky–Talapov model [68] which corresponds to a sine–Gordon model with an additional term controlling the soliton number. The wave vector difference of the Raman laser beams induces a linearly growing potential imbalance between the two condensates, playing the role of the soliton chemical potential, and at sufficiently high strength it induces winding of the relative phase. Due to the presence of boundaries, the quantised solitons that are injected in this way in a finite size system are

expected to form a regular lattice, naturally giving rise to an inhomogeneous structure. In addition, the underlying microscopic model indicates the existence of a coupling between the atomic density and the relative phase which in combination with the DMD functionality may be exploited to control the soliton density profile [103]. Overall the recent experimental and theoretical progress of Ref. [101, 102] and [61] lays the groundwork for the implementation of states with localised soliton density bumps similar to those considered here and the study of their dynamics under the sine–Gordon model.

The relevant parameters of the protocol studied in our work are given by the sine–Gordon coupling $\beta$ which is in one-to-one correspondence with the Luttinger parameter $K$ [58], and the amplitude $A$ and width $\sigma$ of the external field profile. Varying these parameters leads to several interesting physical effects, which we discuss below.

**Ground state transition –** The initial state displays an interesting transition when changing either the amplitude $A$ of the external field or the interaction parameter $\beta$ of the sine–Gordon model. The transition takes place both in the quantum and the classical sine–Gordon field theory and is reflected by changes in the initial profile of the field $\varphi$ and the topological charge density ($\propto \partial_x \varphi$), as well as jumps in the field zero mode. In particular, we showed that fixing the amplitude $A$ of the external field and varying the interaction parameter $\beta$ or the other way round, discontinuous and consecutive transitions happen in the classical theory at special values of $A\beta$.

This phenomenon can be understood from the bounded nature of the cosine potential. As $A\beta$ increases, it becomes energetically favourable to shift the field profiles $\beta\varphi(x)$ by $\pi$ (or multiples of $\pi$), which guarantees that the field can take values close to two vacua of the cosine potential over extended spatial regions. Studying the classical theory, some transition points were precisely determined, which are in remarkably good agreement with our observations from TCSA used to study the quantum theory. Below the first transition point (i.e. for small $\beta$ or $A$), the classical and quantum profiles were found to be very similar to each other, as well as to profiles obtained from a Klein–Gordon theory with the same scalar particle mass. However, beyond the first transition point, the classical and quantum profiles also show some important differences: the latter has features of the analogous free Dirac fermion problem for high enough amplitudes of the initial external field, at least in the investigated parameter regime. We find it important to stress that the above transition is present also in the Dirac theory when changing the amplitude of the external field. Although we have no conclusive evidence, the transition in the quantum theory is expected to be steep but continuous, contrary to what is observed for the classical counterpart.

The emergence of the fermionic features of the inhomogeneous initial states in the interacting quantum sine–Gordon theory, which are absent in the classical case, can be attributed to their enhanced soliton content, since at the quantum level solitons are naturally related to fermionic excitations via the equivalence between the sine–Gordon and massive Thirring models. Nevertheless, as evidenced by the Dirac theory also exhibiting a transition to a Klein–Gordon-like initial state, fermionic excitations such as solitons can also give rise to bosonic behaviour for sufficiently small external fields and hence low particle densities.

**Failure of the local density approximation (LDA) –** Our results show that many features of the quantum inhomogeneous ground states, in particular the observed transition are not captured by the local density approximation, as we explicitly demonstrated at the free fermion point of the sine–Gordon model. In particular, for external fields with small amplitude $A$, the predictions of LDA for the (topological) charge density profiles are significantly different from the result of the exact free fermion computation. As expected, LDA can reproduce faithfully the initial profiles when the amplitude of the external field is large.

**Interpolation between bosonic and fermionic behaviour –** The transition in the inhomogeneous ground state of the quantum theory is only one of the manifestations of a boson-fermion transition. Another one is the crossover in the profile of the time evolving front from one characteristic of free boson dynamics for small $\beta$ to a fermionic behaviour at $\beta^2 = 4\pi$. The smooth interpolation between the two front behaviours can be interpreted in the light of the spectrum of bosonic states in the theory. As the value of $\beta$ increases, the breather particles (corresponding to bound states of the fundamental bosonic excitation) disappear from the spectrum one by one, with the $n$th one vanishing at the threshold $\beta^2 = 8\pi/(n+1)$. Finally, at $\beta^2 = 4\pi$, the fundamental bosonic excitation also disappears from the spectrum, which is signalled by its spectral weight going to zero. As a result, the temporal dynamics shows progressively less sign of the typical oscillatory tails associated with bosonic excitations, and becomes more and more dominated by the solitons, which at $\beta^2 = 4\pi$ eventually turn into free fermions, resulting in narrowly localised fronts moving along the light cone.

However, the crossover in the time evolving profile can be also observed when the interaction parameter is kept fixed and the magnitude of the external field is varied. In particular, this behaviour is also displayed by the free massive Dirac theory and hence cannot be understood by the inherent bosonic and fermionic excitations of the homogeneous theory. In such a scenario, instead, a collective bosonic behaviour of the otherwise fermionic excitations can be held responsible for the Klein–Gordon-like behaviour. This behaviour is triggered by the initial external field via the inhomogeneous initial state, and interestingly such collective excitations seem to be long-lived enough to influence the dynamics at least up to intermediate times, although the corresponding 'bosonic' oscillatory features are clearly suppressed as time progresses. Even though we only treated the attractive regime in this paper, our results lead us to expect that the appearance of such collective bosonic degrees of freedom for low enough densities is likely to be a dominant effect in the repulsive regime in the sine–Gordon model as well. The study of this low density bosonic behaviour in the repulsive regime, and in particular, the temporal suppression of the corresponding bosonic oscillations is an interesting open question for further investigations.

**Experimental implications –** In the classical theory, the transition in the initial state only depends on the combination $\beta A$, therefore increasing $\beta$ can be traded for increasing $A$. However, in the quantum theory $\beta$ becomes an independent parameter governing the size of quantum fluctuations of the scalar field, so this interchangeability is, strictly speaking, no longer valid. It is not clear at the moment whether the transition still occurs for any fixed value for $\beta$ by changing the external field. This question is particularly interesting in the repulsive regime of the quantum sine–Gordon model (hosting only solitonic excitations) especially far away from the free fermion point, and also in the small $\beta$ regime, none of which are presently amenable to TCSA studies; however, at least the latter is accessible for experiments [52]. Finally, we recall the observation from Subsection 4.1 that albeit the interchangeability of $\beta$ and $A$ is not exactly valid at the quantum level we note that even if the experimental realisation is restricted to small $\beta$, strong coupling phenomena can be explored by increasing the amplitude $A$ of the external field.

# Acknowledgements

The authors are grateful to R. Horváth and M. Lájer for their useful pieces of advice which helped us implement the improved numerical method. We are also grateful to K. Hódsági and G. Fehér for their early stage contributions to method development and to O.A. Castro-Alvaredo, J. Schmiedmayer and P. Calabrese for useful discussions. This work was partially

supported by the National Research, Development and Innovation Office (NKFIH) within the Quantum Information National Laboratory of Hungary, and through the OTKA Grants K 138606 and SNN139581. S. S. acknowledges support by the Slovenian Research Agency (ARRS) under grant QTE (Grant No. N1-0109) and by the Foundational Questions Institute (Grant No. FQXi-IAF19-03-S2). D. X. H. acknowledges support from ERC under Consolidator grant number 771536 (NEMO). M. K. acknowledges support by a "Bolyai János" grant of the HAS and by the ÚNKP-21-5 new National Excellence Program of the Ministry for Innovation and Technology from the source of the National Research, Development and Innovation Fund.

# A  Truncated conformal space approach

In this Appendix we provide a brief review of application of the truncated conformal space approach (TCSA) to the inhomogeneous sine–Gordon model. We also discuss the issue of extrapolation and the improvements we introduced in the computation of the initial state and its time evolution.

## A.1  Sine–Gordon model as a perturbed conformal field theory

First of all let us formulate both the homogeneous and inhomogeneous sine–Gordon theory in the PCFT language. Introducing the rescaled quantum field $\hat{\varphi}$ and external source $J(x)$ as

$$\hat{\varphi}(x,t) = \frac{\hat{\phi}(x,t)}{\sqrt{4\pi}}, \qquad j(x) = \frac{J(x)}{\sqrt{4\pi}} \tag{A.1}$$

and the compactification radius $R$ as

$$\beta = \frac{\sqrt{4\pi}}{R}, \tag{A.2}$$

we can rewrite the inhomogeneous quantum Hamiltonian as

$$\begin{aligned}
\hat{H}_{\text{inhom}} =& \hat{H}_{\text{sG}} + \hat{H}_j \\
=& \frac{1}{4\pi} \int_{-L/2}^{L/2} \mathrm{d}x \left\{ \frac{1}{2} \hat{\pi}(x,t)^2 + \frac{1}{2} \left( \partial_x \hat{\phi}(x,t) \right)^2 - \lambda \cos\left( \frac{\hat{\phi}(x,t)}{R} \right) \right\} \\
& - \frac{1}{4\pi} \int_{-L/2}^{L/2} \mathrm{d}x \, \partial_x \hat{\phi}(x,t) J'(x),
\end{aligned} \tag{A.3}$$

where the last term is the inhomogeneous part $\hat{H}_j$ and the rest is the homogeneous sine–Gordon Hamiltonian $\hat{H}_{\text{sG}}$. We recall that $J(x)$ is a classical and static external field compatible with periodic boundary conditions, that is $J(-L/2) = J(L/2)$ and $J'(x)$ denotes its spatial derivative. The coupling constant $\lambda$ is related to the first breather mass $m_1$ as [79]

$$\lambda = \left( 2\sin\frac{\xi\pi}{2} \right)^{2\Delta-2} \frac{2\Gamma(\Delta)}{\pi\Gamma(1-\Delta)} \left( \frac{\sqrt{\pi}\Gamma\left(\frac{1}{2-2\Delta}\right) m_1}{2\Gamma\left(\frac{\Delta}{2-2\Delta}\right)} \right)^{2-2\Delta} \qquad \text{with} \qquad \Delta = \frac{\beta^2}{8\pi}. \tag{A.4}$$

Imposing periodic boundary conditions, the compactified quantum field $\hat{\phi} \equiv \hat{\phi} + 2\pi kR$ admits the following mode expansion

$$\hat{\phi}(x,t) = \hat{\phi}_0 + \frac{4\pi}{L}\hat{\pi}_0 t + \frac{4\pi}{L}\frac{\hat{M}xR}{2} + i\sum_{k\neq 0}\frac{1}{k}\left[ a_k \exp\left( i\frac{2\pi}{L}k(x-t) \right) + \bar{a}_k \exp\left( -i\frac{2\pi}{L}k(x+t) \right) \right], \tag{A.5}$$

where the winding number operator $\hat{M}$ has integer eigenvalues and corresponds to the topological charge $\hat{Q}$. The operators $\hat{\phi}_0$ and $\hat{\pi}_0$ are the zero modes of the field $\hat{\phi}$ and its conjugate momentum field $\hat{\pi} = \partial_t \hat{\phi}$ and the $a_k$ and $\bar{a}_k$ correspond to right and left oscillator modes creating/annihilating particles with momenta $p = \pm 2\pi|k|/L$. These operators satisfy

$$
\begin{aligned}
[\hat{\phi}_0, \hat{\pi}_0] &= i\,, \\
[a_k, a_l] = [\bar{a}_k, \bar{a}_l] &= k\delta_{k+l}\,,
\end{aligned}
\tag{A.6}
$$

that is, $a_k$ and $\bar{a}_k$ with negative/positive $k$ can be interpreted as creation/annihilation operators.

The homogeneous sine–Gordon Hamiltonian $\hat{H}_{\mathrm{sG}}$ can be rewritten as

$$
\hat{H}_{\mathrm{sG}} = \frac{1}{4\pi} \int_{-L/2}^{L/2} dx \left\{ \frac{1}{2} : \hat{\pi}^2 + \left(\partial_x \hat{\phi}\right)^2 : -\frac{\lambda}{2}\left(V_n^{\mathrm{cyl}} + V_n^{\mathrm{cyl}}\right) \right\}
\tag{A.7}
$$

in terms of the vertex operators

$$
V_n^{\mathrm{cyl}} = : e^{in\hat{\phi}/R} :^{\mathrm{cyl}}\,,
\tag{A.8}
$$

where the semicolon denotes normal ordering with respect to the massless scalar field modes. The space-time geometry of the model corresponds to a cylinder and the use of upper index "cyl" of the normal ordering indicates that these vertex operators have the standard CFT normalisation specified below in (A.9). It is useful to analytically continue to imaginary time $\tau = -it$, and introduce complex coordinates $w = \tau - ix$, $\bar{w} = \tau + ix$ on the resulting Euclidean space-time cylinder. The above mentioned normalisation of the vertex operators is defined by the short distance behaviour of their two-point functions:

$$
\langle 0|V_n^{\mathrm{cyl}}(w_1, \bar{w}_1) V_m^{\mathrm{cyl}}(w_2, \bar{w}_2)|0\rangle = \frac{\delta_{n,-m}}{|w_1 - w_2|^{4n^2\Delta}} + \text{subleading terms}\,.
\tag{A.9}
$$

To compute matrix elements of the vertex operators, it is useful to perform a conformal transformation $z = \exp\frac{2\pi}{L}w$, which maps the cylinder to the complex plane parameterised by the dimensionless complex coordinates $z$ and $\bar{z}$, under which the vertex operators transform as [81]

$$
V_{\pm 1}^{\mathrm{pl}}(z, \bar{z})\left(|z|\frac{2\pi}{L}\right)^{2\Delta} = V_{\pm 1}^{\mathrm{cyl}}(w, \bar{w})\,.
\tag{A.10}
$$

This allows us to express (A.7) as

$$
\hat{H}_{\mathrm{sG}} = \hat{H}_{\mathrm{CFT}} - \lambda\left(\frac{2\pi}{L}\right)^{2\Delta}\frac{L}{2}\left(V_{+1}^{\mathrm{pl}}(0,0) + V_{-1}^{\mathrm{pl}}(0,0)\right)\delta_{q,q'}\,,
\tag{A.11}
$$

where $\delta_{q,q'}$ indicates that action of the vertex operators is restricted to states with the same total momentum, due to the spatial integration in $\hat{H}_{\mathrm{sG}}$. The first term of (A.11) is the conformal Hamiltonian

$$
\hat{H}_{\mathrm{CFT}} = \frac{2\pi}{L}\left(\hat{\pi}_0^2 + \frac{M^2}{8\pi\beta^2} + \sum_{k>0} a_{-k}a_k + \sum_{k>0} \bar{a}_{-k}\bar{a}_k - \frac{1}{12}\right)\,.
\tag{A.12}
$$

The inhomogeneous term can be expressed using the Fourier representation of the current $J(x)$

$$
J(x) = \sum_{n=-\infty}^{\infty} e^{-i\frac{2\pi n}{L}x} J_n\,, \qquad J_n = \frac{1}{L}\int_{-L/2}^{L/2} dx\, e^{i\frac{2\pi n}{L}x} J(x)\,,
\tag{A.13}
$$

as

$$\hat{H}_j = -\frac{2\pi}{L}\left(\frac{i}{2}\right)\sum_{k\neq 0} k\left(J_k a_k + J_{-k}\bar{a}_k\right), \tag{A.14}$$

where we assumed

$$\int_{-L/2}^{L/2} \mathrm{d}x\, J'(x) = 0. \tag{A.15}$$

## A.2 The Hilbert space and its truncation

The Hilbert space $\mathcal{H}$ of the compactified boson is composed of Fock modules $\mathcal{F}_{n,M}$, built upon Fock vacua $|n,M\rangle$ using the oscillator modes, where the states $|n,M\rangle$ $(n,M \in \mathbb{Z})$ have eigenvalues $n/R$ under $\hat{\pi}_0$ and $M$ under the winding number operator $\hat{M}$. The vertex operators defined in the previous subsection $V_n$ correspond to $V_{n,0}$ and do not connect modules with different winding number. Using the Fock decomposition of the free boson Hilbert space $\mathcal{H}$ we can write

$$\mathcal{H} = \bigoplus_{n,M} \mathcal{F}_{n,M}, \tag{A.16}$$

with $n,M \in \mathbb{Z}$ where each Fock module is spanned by the vectors

$$a_{-k_1}...a_{-k_r}\bar{a}_{-p_1}...\bar{a}_{-p_l}|n,M\rangle : r,l \in \mathbb{N}, k_i, p_j \in \frac{2\pi}{L}\mathbb{N}^+, \tag{A.17}$$

which are eigenstates of $\hat{H}_{\text{CFT}}$ with energy

$$E = \frac{2\pi}{L}\left(\frac{(n\beta)^2}{4\pi} + \frac{M^2}{8\pi\beta^2} + \sum_{i=1}^{r}k_i + \sum_{j=1}^{l}p_j - \frac{1}{12}\right). \tag{A.18}$$

The fact that (A.14) does not depend on $M$ due to the neutrality condition of the external field (A.15), has important consequences. As well-known, $[\hat{H}_{\text{sG}},\hat{Q}] = 0$ and the ground state of the homogeneous quantum sine–Gordon model is in the $\hat{M} = 0$ sector. From (A.14) it is very easy to show that also $[\hat{H}_j,\hat{Q}] = 0$ holds. Since $\hat{H}_j$ acts exactly the same way on each subspace of the Hilbert space with a specific winding number $M$, and for non-zero $M$ these subspaces have an additional energy $M^2/(8\pi\beta^2)$ according to (A.18), one can deduce that the ground state of the inhomogeneous problem with $\hat{H}_{\text{inhom}}$ lives in the $M = 0$ sector as well. Therefore in the following, we can restrict ourselves to Fock modules $\mathcal{F}_{n,0}$ which we denote merely as $\mathcal{F}_n$ for simplicity and dropping index $M$ from now on.

It is useful to further decompose the Hilbert space into different momentum sectors as well according to

$$\mathcal{H} = \bigoplus_{n,q} \mathcal{F}_n^{(q)}, \tag{A.19}$$

where $\mathcal{F}_n^{(q)}$ denotes the momentum $q$ subspace of the $n$th Fock module spanned by the vectors

$$a_{-k_1}...a_{-k_r}\bar{a}_{-p_1}...\bar{a}_{-p_l}|n\rangle, \quad r,l \in \mathbb{N}, \quad k_i, p_j \in \frac{2\pi}{L}\mathbb{N}^+, \quad \sum k_i - \sum p_j = q. \tag{A.20}$$

In our numerical studies we used the simplest and most common truncation scheme, when the truncation criterion is the energy[2]. In this case we keep states in the truncated conformal Hilbert space whose energy does not exceed $2\pi e_c/L$, and so the truncated space is given by

$$\mathcal{H}_{\text{TCSA}}(e_c) = \text{span}\left\{a_{-k_1}...a_{-k_r}\bar{a}_{-p_1}...\bar{a}_{-p_l}|n\rangle : \frac{(n\beta)^2}{4\pi} + \sum_{i=1}^{r}k_i + \sum_{j=1}^{l}p_j - \frac{1}{12} \le e_c\right\}. \tag{A.21}$$

---

[2]Other truncation schemes can be applied as well depending on the problem considered [58].

Turning now to the physical operators the matrix element of the operators $a_k$ and $\bar{a}_k$, can be straightforwardly computed. These operators act between momentum subspaces of the Fock modules according to

$$
\begin{aligned}
a_k &: \mathcal{F}_n^{(q)} \to \mathcal{F}_n^{(q+k)}, \\
\bar{a}_k &: \mathcal{F}_n^{(q)} \to \mathcal{F}_n^{(q-k)}.
\end{aligned}
\tag{A.22}
$$

Matrix elements of the vertex operators $V_m^{\mathrm{pl}}$ can be computed in the conformal basis using the mode expansion of the canonical field $\phi$ (A.5). Combined with the $\delta_{q,q'}$ factors in (A.11), these operators act as

$$
\begin{aligned}
V_{+a}^{\mathrm{pl}}(0,0)\delta_{q,q'} &: \mathcal{F}_n^{(q)} \to \mathcal{F}_{n+a}^{(q)}, \\
V_{-a}^{\mathrm{pl}}(0,0)\delta_{q,q'} &: \mathcal{F}_n^{(q)} \to \mathcal{F}_{n-a}^{(q)}.
\end{aligned}
\tag{A.23}
$$

The standard sine–Gordon TCSA has been implemented in numerous instances with various purposes including the investigation of equilibrium [70,71] and out-of-equilibrium properties of the model [58,75,76]. To perform the computations shown in the present work, we used a recently implemented improved algorithm which uses the left/right factorisation of conformal field theory under periodic boundary conditions. The details of this implementation have been published in [104].

## A.3 Cut-off dependence and extrapolation in TCSA

An important consequence of the truncation is that all quantities computed from TCSA possess a cut-off dependence, and physical results can be recovered in the limit when the cut-off is removed. This cut-off dependence can be addressed using renormalisation group methods [82–86]. In particular, vacuum expectation values of local operators possess a leading order power-law dependence on the energy cut-off and consequently, the infinite cut-off expression can be estimated by numerical extrapolation. Avoiding any technical details, we briefly present the most important results for our purposes. A more detailed summary on this subject can be found in Ref. [58] and a rigorous treatment in Ref. [87].

Let us first introduce $n$ to describe the energy cut-off as

$$
n = \frac{e_c}{2}
\tag{A.24}
$$

and denote the expectation value of the operator $\mathcal{O}$ at a cut-off $n$ by $\langle \mathcal{O} \rangle^{(n)}$. It was shown in [87] that the leading cut-off dependence can be written as

$$
\langle \mathcal{O} \rangle^{(n)} = \langle \mathcal{O} \rangle^{(\infty)} + \sum_A K_A n^{2\alpha_A - 2} \left( 1 + O\left(\frac{1}{n}\right) \right),
\tag{A.25}
$$

where $\langle \mathcal{O} \rangle^{(\infty)}$ is the expectation value with the cut-off removed. Using this relation, data points obtained for a sequence of sufficiently high $n$ can be extrapolated numerically to obtain a precise estimate for the expectation value $\langle \mathcal{O} \rangle^{(\infty)}$. The exponents $\alpha_A$ in (A.25) can be analytically determined via the operator product expansion of the perturbing field with the observable $\mathcal{O}$. In particular, for the operator $\partial_x \phi$ and perturbing field $(V_{+1} + V_{-1})/2$ the leading exponent $\alpha_A$ turns out to be zero [58] and so

$$
\langle \partial_x \hat{\phi} \rangle^{(n)} = \langle \partial_x \hat{\phi} \rangle^{(\infty)} + K n^{-2} \left( 1 + O\left(\frac{1}{n}\right) \right).
\tag{A.26}
$$

As shown in Ref. [87], the leading order cut-off extrapolation can be used in excited states as well, as long as the cut-off is large enough compared to the energy of the excited state under

consideration. It is, nevertheless, not entirely obvious from [87] whether this method can be applied for expectation values in inhomogeneous initial states and especially in states subject to time evolution. Concerning first the expectation values in inhomogeneous states, the validity of extrapolation procedure is strongly motivated by simple physical arguments: as long as the typical length scale $l$ at which $\langle \partial_x \hat{\phi}(x) \rangle$ (or another operator in under investigation) varies is larger than the inverse mass gap, (A.26) is expected to give accurate predictions. Giving rigorous justifications for the validity of (A.26) under non-trivial time evolution is rather difficult. Extrapolation in time-dependent scenarios was proven to be reliable in some particular settings [58, 72].

In this work we use extrapolation primarily for the study of the initial state, where its applicability is more justified. For time evolving quantities we instead present the data corresponding to the highest possible cut-off, but as we demonstrate in Appendix A.5, the behaviour of the extrapolated time evolved expectation values are qualitatively similar to the results obtained at the highest value of the cut-off.

### A.4 Improvements for the computation of the initial state and of the time evolution

In the conventional TCSA one usually explicitly stores each (non-zero) matrix element of operators in the computer's memory, which then becomes the main limitation of the method as the required memory scales with the square of the dimension of the truncated Hilbert space. In our improved method, we exploit the fact that these matrices can be built from much smaller blocks, and their explicit construction is not necessary for many purposes. This idea was first implemented in [74].

In particular, the action of $V^{\text{pl}}_{\pm n}, \hat{H}_{\text{sG}}, a_k, \bar{a}_k$ as well as of $\hat{H}_{\text{inhom}}$ on generic vectors can be easily computed without the explicit construction of the corresponding matrices on the truncated Hilbert space. As a consequence, expectation values of the above operators are straightforward to obtain.

To determine the initial state $|0\rangle_j$ and to compute the time evolved vector $e^{-it\hat{H}_{\text{sG}}}|0\rangle_j$, it is sufficient to specify the actions of $\hat{H}_{\text{sG}}$ and $\hat{H}_{\text{inhom}}$ on arbitrary vectors. The numerical determination of the ground state $|0\rangle_j$ of $\hat{H}_{\text{inhom}}$ can be achieved by improved Krylov subspace methods that require only the action of the matrix. Such methods are widely implemented; here we used the built-in `eigs` eigensolver of Matlab [105] to compute the lowest energy states of $\hat{H}_{\text{inhom}}$.

The time evolved state $e^{-it\hat{H}_{\text{sG}}}|0\rangle_j$ was computed using a simple numerical implementation of the Chebyshev method [72]. To evaluate the time evolution of the system starting from $|0\rangle_j$, the time evolution operator $e^{-i\hat{H}_{\text{sG}}t}$ is expanded on the basis which are known to give the best approximation of the exponential to any finite order. The Chebyshev polynomials $T_n(x)$ are defined by the recurrence relation

$$T_{n+1}(x) = 2x T_n(x) - T_{n-1}(x) \quad , \quad T_0(x) = 1 \quad , \quad T_1(x) = x \tag{A.27}$$

and form a complete basis for functions on the interval $[-1, 1]$. The exponential time evolution operator can be expanded as

$$e^{-i\hat{H}_{\text{sG}}t}|0\rangle_j = J_0(at)|0\rangle_j + 2\sum_{n=1}^{\infty} (-i)^n J_n(at)|0\rangle_j^{(n)}, \tag{A.28}$$

$$|0\rangle_j^{(n)} = T_n(\hat{H}_{\text{sG}}/a)|0\rangle_j,$$

with $|0\rangle_j^{(0)} = |0\rangle_j$ using the Bessel functions

$$J_n(z) = \sum_{l=0}^{\infty} \frac{(-1)^l}{l!(k+l)!} \left(\frac{z}{2}\right)^{2l+k} , \tag{A.29}$$

where $a$ is a real number which is larger than the modulus of all eigenvalues of the (truncated) Hamiltonian $\hat{H}_{\text{sG}}$, but otherwise arbitrary. The expansion can be truncated at an appropriate order to get an approximation for the time evolution operator; it turns out that it is necessary to truncate at a level $n_{max} \gtrsim a t_{max}$, with $t_{max}$ the time we aim to reach (which is usually limited by the finite volume and light speed anyway). To use (A.28) the necessary vectors can be computed recursively

$$|0\rangle_j^{(1)} = \frac{1}{a}\hat{H}_{\text{sG}}|0\rangle_j^{(0)} ,$$
$$|0\rangle_j^{(n+1)} = 2\frac{1}{a}\hat{H}_{\text{sG}}|0\rangle_j^{(n)} - |0\rangle_j^{(n-1)} ,$$

for which only the matrix action is needed.

Using the improved version of sine–Gordon TCSA, the dimension of the Hilbert space can be substantially increased compared to the conventional method. Whereas with the standard TCSA the typical size of the Hilbert space is of the order $10^4$, with the improved method one can easily reach the order of $10^6 - 10^7$. Such a large Hilbert space is in fact necessary for our purposes. While for the time evolution each momentum sector of the Hilbert space can be treated independently (as $\hat{H}_{\text{sG}}$ is a translationally invariant Hamiltonian), this factorisation is not present when an inhomogeneous source term is added which is the case when computing the inhomogeneous initial state.

Finally we note that the classical inhomogeneous initial state (2.19) as well as its time evolution (2.11) were computed using the NDSolve routine of Wolfram Mathematica [106].

### A.5 Extrapolated dynamical quantities

Here we present the dynamical evolution of the QFT topological density profiles using cut-off extrapolation, explained in detail in Appendix A.3. We stress again that using the extrapolation procedure is only justified for equilibrium quantities, although there are known cases where its good performance in out-of-equilibrium scenarios was demonstrated [58]. The applicability of extrapolation to the following setting is therefore not clear, nevertheless it is instructive to demonstrate the corresponding results. Figures A.1 and A.2 correspond to fixing the amplitude $A$ of the external field and changing the interaction parameter $\beta$ and the vice versa, respectively. The main message of these figures is that extrapolation introduces only slight quantitative differences compared to the simulations performed at the highest cut-offs. In fact, the magnitude of various spatial structures (bumps and dips) in the density profiles are usually increased by the procedure. Extrapolation may also introduce unphysical features, such as high frequency oscillations at late times, which can be seen in the top right and left corners of some of the subplots of Fig. A.2. For $\beta_{\text{FF}}A/m_1 = 15, m_1\sigma = 4/9$ and $\beta = \beta_{\text{FF}}/1.7$; and $\beta_{\text{FF}}A/m_1 = 12, m_1\sigma = 1/3$ and $\beta = \beta_{\text{FF}}/\sqrt{2}$ and $\beta_{\text{FF}}/1.5$, extrapolation was carried out in the way explained in Appendix B.1, due to hybridisation. For the latter case, the ground state in the $e_c = 28$ Hilbert space was embedded to the $e_c = 30$ Hilbert space as explained in Appendix B.1.

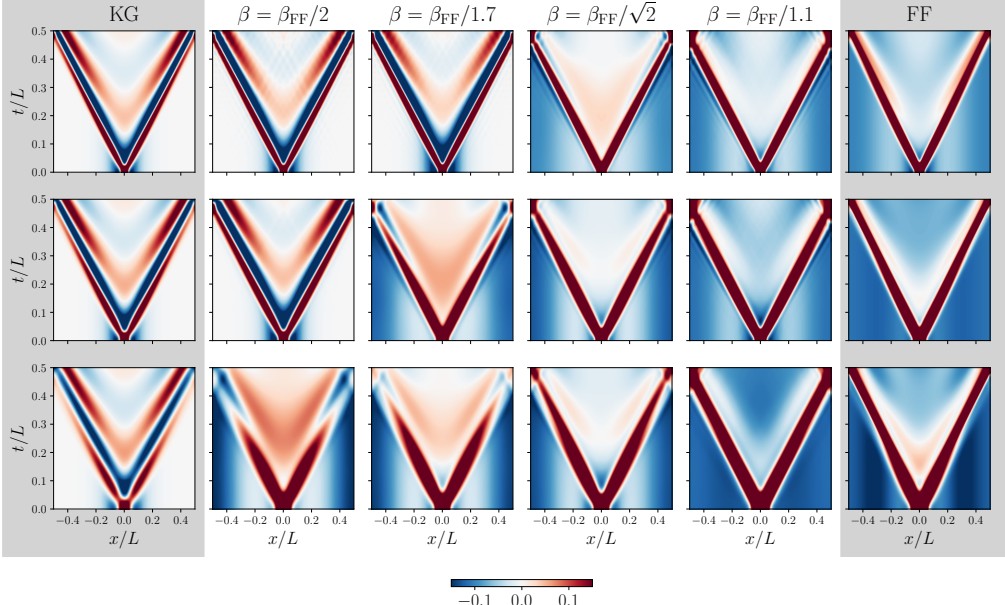

Figure A.1: **Transition from free boson-like to free fermion-like behaviour after a local quantum quench in the sine–Gordon model:** Density plots of the time evolution of the soliton density $\rho(x,t) = \beta \partial_x \varphi(x,t)/(2\pi)$ as a function of the space and time coordinates $x, t$ at various values of the parameters. From left to right the sine–Gordon coupling $\beta$ changes from 0 (Klein–Gordon limit) to $\beta_{\mathrm{FF}} = \sqrt{4\pi}$ (free fermion point), while from top to bottom the initial external field bump changes from shorter and thinner to taller and wider bumps (the three rows correspond to the bump height parameter $A\beta_{\mathrm{FF}}/m_1$ and the Gaussian width parameter $m_1\sigma$ taking the values $(12, 1/3), (15, 4/9)$ and $(18, 2/3)$, respectively). Note the change in the propagation fronts from having oscillatory tails (Klein–Gordon dynamics) to fast decaying tails (free fermion dynamics). Note also that the background is neutral in the free boson limit, while at the free fermion point a negatively charged background appears to compensate for the positive bump charge. This change is triggered by a crossover transition in the initial state, which occurs at an intermediate value of $\beta$ which depends on the initial bump strength. Compared to Fig. 1.1 of the main text, the results shown here were extrapolated using data with cut-offs $e_c = 24, 26, 28$ and 30.

# B  Inhomogeneous initial state

## B.1  Eigenstate crossing and hybridisation

The transition of the inhomogeneous ground state, discussed in Subsection 4.1 of the main text, is observed in both the quantum and the classical theory when varying either the interaction strength $\beta$ or the amplitude $A$ of the external inhomogeneous field.

However, in the quantum theory this transition can also be observed for fixed $\beta$ and $A$ by changing the upper energy cut-off parameter $e_c$, which controls the truncation of the Hilbert space in TCSA. Although the energy cut-off $e_c$ may appear as a less physical variable than $A$ or $\beta$, regarding it as a control parameter provides another point of view on the physical phenomenon. Perhaps more importantly, at least from a technical point of view, the transition as a function of $e_c$ induces some slight difficulties for TCSA extrapolation, making its brief discussion necessary.

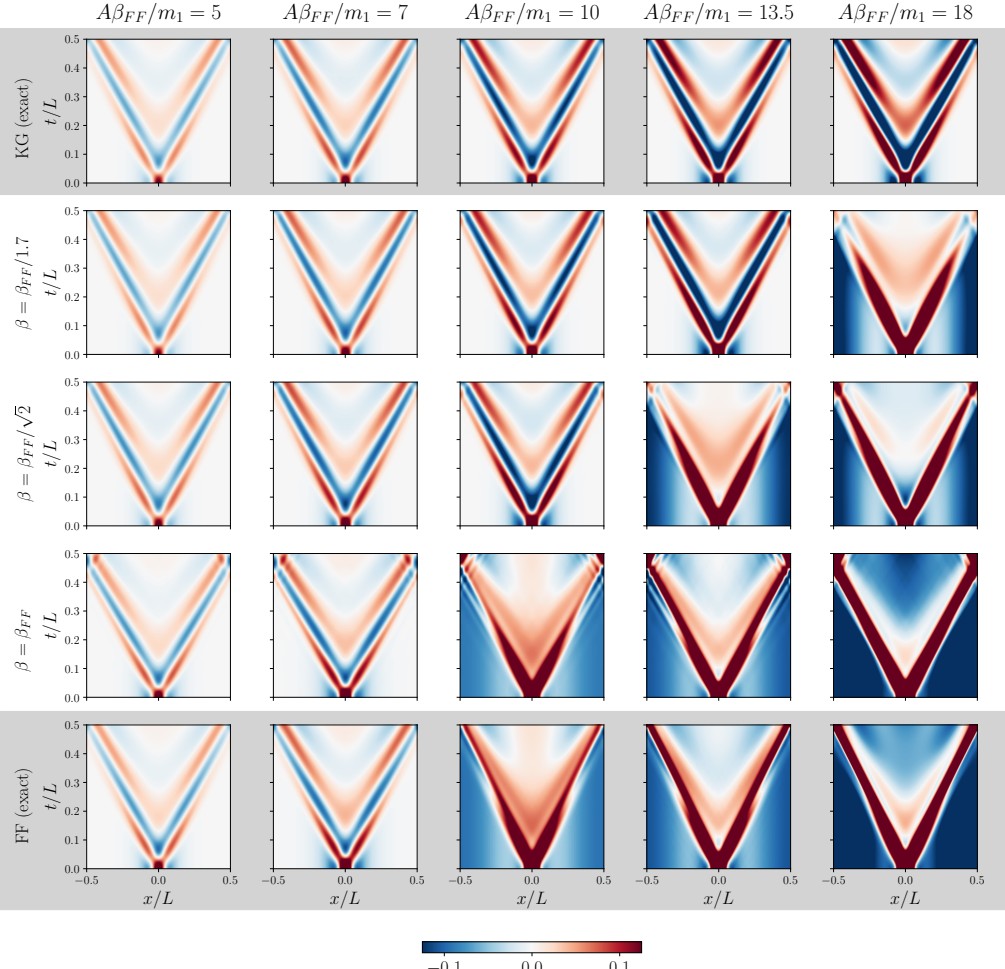

Figure A.2: The time evolved QFT expectation value of $m_1^{-1}\langle\hat{\rho}(x,t)\rangle_j$ for five differ­ent amplitudes of the external field with bump width bump-width $m_1\sigma = 2/3$. The interaction parameters are $\beta = 0$ (Klein–Gordon limit) $\beta = \beta_{\text{FF}}/1.7$, $\beta = \beta_{\text{FF}}/\sqrt{2}$ and $\beta = \beta_{\text{FF}}$. For the computation of the free fermion dynamics, both TCSA and the nu­merically exact method was used. From the Klein–Gordon dynamics, $m_1^{-1}\langle\hat{\rho}(x,t)\rangle_j$ was obtained by rescaling $m_1^{-1}\langle\partial_x\hat{\varphi}(x,t)\rangle_j$ with $1/ = 1.7\sqrt{\pi}$) which corresponds to the $\beta = \beta_{\text{FF}}/1.7$ point. For the TCSA quantities, extrapolation was used based on data with cut-offs $e_c = 24, 26, 28$ and $30$.

The initial state transition occurs in the quantum model when the interaction $\beta$ and the amplitude of the external field $A$ are chosen to be close the 1st transition point in the clas­sical theory. As discussed in Subsection 4.1 of the main text, for the three different Gaussian external fields $j'(x)$ with widths $m_1\sigma = 2/3, 4/9$ and $1/3$, the first transition occurs classic­ally at $A\beta/m_1 = 8.8, 7.856$ and $7.42$, respectively, and when the amplitudes are fixed corres­pondingly as $A\beta_{\text{FF}}/m_1 = 18, 15$ and $12$, the transition values for $\beta$ are $\beta_{\text{FF}}/2.045, \beta_{\text{FF}}/1.909$ and $\beta_{\text{FF}}/1.617$, respectively. In particular the initial state transition is the most articu­lately shown by the quantum theory when $\beta = \beta_{\text{FF}}/2$ for $(A\beta_{\text{FF}}/m_1, m_1\sigma) = (18, 2/3)$, $\beta = \beta_{\text{FF}}/1.7$ for $(A\beta_{\text{FF}}/m_1, m_1\sigma) = (15, 4/9)$ and $\beta = \beta_{\text{FF}}/1.5$ as well as $\beta = \beta_{\text{FF}}/\sqrt{2}$ for $(A\beta_{\text{FF}}/m_1, m_1\sigma) = (12, 1/3)$.

In all these cases the initial state transition in the QFT can be summarised as follows: increasing the energy cut-off $e_c$, the energies of the first two eigenstates of the inhomogeneous quantum Hamiltonian $\hat{H}_{\text{inhom}}$ approach each other or even cross. When the energy difference

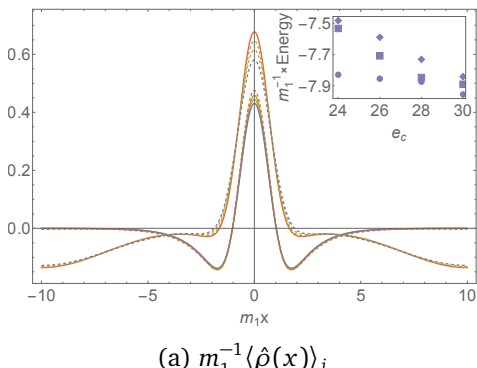
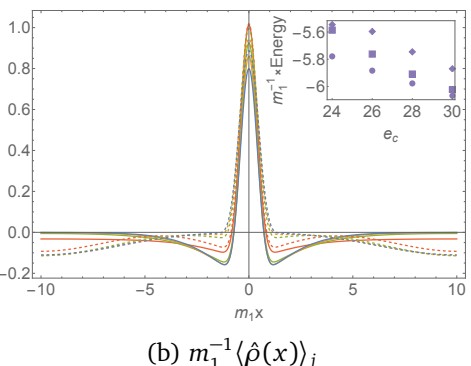

(a) $m_1^{-1}\langle\hat{\rho}(x)\rangle_j$                      (b) $m_1^{-1}\langle\hat{\rho}(x)\rangle_j$

Figure B.1: Eigenstate crossing (a) and hybridisation (b) in the inhomogeneous quantum sine–Gordon problem w.r.t. changing the truncation cut-off. The figures display the QFT expectation value of the topological charge density $\langle\hat{\rho}(x)\rangle_j$ in the ground state (continuous lines) and the first excited state (dashed lines). The parameters are $m_1$, $\ell = m_1 L = 20$ and (a) $\beta = \beta_{\text{FF}}/2$, $A\beta_{\text{FF}}/m_1 = 18$, $m_1\sigma = 2/3$ and (b) $\beta = \beta_{\text{FF}}/1.5$, $A\beta_{\text{FF}}/m_1 = 12$, $m_1\sigma = 1/3$. The blue, orange green and red curves correspond to cut-offs 24, 26, 28 and 30. The insets display the energies of the first three eigenstates.

of the two eigenstates is not too small, the one that has the lowest energy for higher cut-offs typically admits field profiles similar to the classical ones observed after the 1st transition. The field expectation value in the other eigenstate, instead, typically looks similar to that of the Klein–Gordon theory.

In the clearest case $(m_1\beta_{\text{FF}}A, m_1\sigma) = (18, 2/3)$, a pure eigenstate transition occurs at $\beta = \beta_{\text{FF}}/2$ as shown in Fig. B.1 (a). The field profiles of $\hat{\rho}$ (and accordingly the quantum states) can be organised into two families: the ground state at $e_c = 30$ and the first excited states at $e_c = 24, 26, 28$ making up one family, and the first excited state at $e_c = 30$ and the ground states at $e_c = 24, 26, 28$ making up the other. Within one family, the profiles look almost exactly the same. Consequently, the ground state at $e_c = 30$ and the first excited states at $e_c = 24, 26, 28$ must be considered when performing cut-off extrapolation for the initial profile, and similarly for time evolving quantities.

The situation is more subtle for the other cases with $\beta = \beta_{\text{FF}}/1.7$ and $(A\beta_{\text{FF}}/m_1, m_1\sigma) = (15, 4/9)$, and $\beta = \beta_{\text{FF}}/1.5$ and $\beta = \beta_{\text{FF}}/\sqrt{2}$ with $(A\beta_{\text{FF}}/m_1, m_1\sigma) = (12, 1/3)$. In these cases one cannot talk about a pure eigenstate transition. The two families of eigenstates can be clearly distinguished, but at a certain cut-off where the difference between the first two eigenstates is small (though not the smallest in all cases), the first and second eigenstates show features of both families. In other words hybridisation occurs which we demonstrate in Fig. B.1 (b) for $\beta = \beta_{\text{FF}}/1.5$ with $(A\beta_{\text{FF}}/m_1, m_1\sigma) = (12, 1/3)$, when the hybridisation affects the data with the highest achievable cut-off $e_c = 30$. In the other two cases $\beta = \beta_{\text{FF}}/1.7$ with $(A\beta_{\text{FF}}/m_1, m_1\sigma) = (15, 4/9)$, and $\beta = \beta_{\text{FF}}/\sqrt{2}$ with $(A\beta_{\text{FF}}/m_1, m_1\sigma) = (12, 1/3)$, the hybridisation occurs at $e_c = 24$. Clearly, extrapolation cannot be naively used when hybridisation is present.

For the case when this phenomenon occurs at an $e_c$ smaller than the highest one available, we used the initial state at the highest cut-off ($e_c = 30$), and truncated this state to the smaller Hilbert space corresponding to lower cut-offs ($e_c = 24, 26, 28$). This way we obtained a set of initial states for which it was possible to extrapolate the corresponding profiles, as well as the time evolved ones once these initial states are subject to time evolution.

An additional subtlety occurs when the hybridisation takes place at the highest reachable cut-off ($e_c = 30$) which is the case for $\beta = \beta_{\text{FF}}/1.5$ and $(A\beta_{\text{FF}}/m_1, m_1\sigma) = (12, 1/3)$. When

Table B.1: Energies of the two configurations $\varphi_j(x)$ with $\beta\varphi_0 = 0$ and $\pi$ when $m\sigma = 4/9$ and $\ell = mL = 20$.

| $\beta A/m$ | 4 | 6 | 7.856 | 8 | 10 | 12 | 14 | 14.412 | 16 |
|---|---|---|---|---|---|---|---|---|---|
| $H_{\text{i.h.}}[\varphi_j(x)]/L,\ \varphi_0 = 0$ | -1.213 | -1.297 | -1.522 | -1.543 | -1.889 | -2.429 | -3.265 | -3.446 | -4.173 |
| $H_{\text{i.h.}}[\varphi_j(x)]/L,\ \varphi_0 = \pi/\beta$ | -0.733 | -1.111 | -1.522 | -1.556 | -2.068 | -2.648 | -3.301 | -3.446 | -4.043 |

Table B.2: The zero mode in the quantum inhomogeneous state from TCSA with $e_c = 30$ and $m_1 L = 20$. $\langle\mathcal{O}\rangle_j$ denotes the expectation value of operator $\mathcal{O}$ in the inhomogeneous initial state, while $\sigma\left[\langle\mathcal{O}\rangle_j\right]$ is the standard deviation characterising the fluctuations.

| $A\beta_{\text{FF}}/m_1 = 15$ $m_1\sigma = 4/9$ | $\langle\cos\beta\hat{\varphi}_0\rangle_j$ | $\langle\cos^2\beta\hat{\varphi}_0\rangle_j$ | $\sigma\left[\langle\cos\beta\hat{\varphi}_0\rangle_j\right]$ | $\langle\sin\beta\hat{\varphi}_0\rangle_j$ | $\langle\sin^2\beta\hat{\varphi}_0\rangle_j$ | $\sigma\left[\langle\sin\beta\hat{\varphi}_0\rangle_j\right]$ | $\beta\varphi_0$ |
|---|---|---|---|---|---|---|---|
| $\beta = \beta_{\text{FF}}/2.3$ | 0.9660 | 0.9355 | 0.0482 | 0 | 0.0640 | 0.2529 | 0 |
| $\beta = \beta_{\text{FF}}/2.0$ | 0.9498 | 0.9073 | 0.0721 | 0 | 0.0919 | 0.3032 | 0 |
| $\beta = \beta_{\text{FF}}/1.7$ | -0.6152 | 0.5628 | 0.4292 | 0 | 0.4338 | 0.6586 | $\pi$ |
| $\beta = \beta_{\text{FF}}/\sqrt{5/2}$ | -0.6274 | 0.5643 | 0.4132 | 0 | 0.4325 | 0.6576 | $\pi$ |
| $\beta = \beta_{\text{FF}}/1.5$ | -0.6224 | 0.5614 | 0.4171 | 0 | 0.4355 | 0.6599 | $\pi$ |
| $\beta = \beta_{\text{FF}}/\sqrt{2}$ | -0.6110 | 0.5559 | 0.4273 | 0 | 0.4409 | 0.6640 | $\pi$ |
| $\beta = \beta_{\text{FF}}/1.1$ | -0.4874 | 0.5192 | 0.5307 | 0 | 0.4751 | 0.6893 | 0 |
| $\beta = \beta_{\text{FF}}$ | -0.3982 | 0.5055 | 0.5890 | 0 | 0.4870 | 0.6979 | 0 |

presenting the simulations corresponding to the highest cut-off at this case, it is more justified to use the $e_c = 28$ simulations, since at this cut-off the field profiles in the first two eigenstates do not mix. It is nevertheless, not obvious, whether the first or the second eigenstate to use as an initial state. On the one hand one can insist on the lowest energy state, on the other hand the dependence of the energies on the cut-off may imply the relevance of the second eigenstate.

## B.2 Transition of the inhomogeneous initial state for additional profiles

In Fig. B.2 we demonstrate the phenomenon of the ground state transition of the inhomogeneous system described by Eq. (2.17) in the main text when the height and the Gaussian width of the external field are $\beta_{\text{FF}}A/m_1 = 15$, $m_1\sigma = 4/9$ and $\beta_{\text{FF}}A/m_1 = 12$ and $1/3$, keeping the amplitudes fixed and varying the interaction parameter $\beta$. Similarly to Sec. 4.1, we compare the quantum and classical initial profiles for $\varphi$ and $\rho \propto \partial_x\varphi$, present the (expectation) values of the zero mode confirming the transition and also determine the first transition points in the classical inhomogeneous problem. For the quantum profiles obtained for $\beta_{\text{FF}}A/m_1 = 15, m_1\sigma = 4/9, \beta = \beta_{\text{FF}}/1.7$ and $\beta_{\text{FF}}A/m_1 = 12, m_1\sigma = 1/3, \beta = \beta_{\text{FF}}/\sqrt{2}$, cut-off extrapolation was carried out in the way explained in Appendix B.1 due to hybridisation.

Following the discussion of Sec. 4.1, we present the energies of the two levels involved in the transitions in Tables B.1 and B.3, while the values of the zero modes are shown in Tables B.2 and B.4.

Table B.3: Energies of the two configurations $\varphi_j(x)$ with $\beta\varphi_0 = 0$ and $\pi$ when $m\sigma = 1/3$ and $\ell = mL = 20$.

| $\beta A/m$ | 4 | 6 | 7.42 | 8 | 10 | 12 | 13.905 | 14 | 16 |
|---|---|---|---|---|---|---|---|---|---|
| $H_{\text{i.h.}}[\varphi_j(x)]/L,\ \varphi_0 = 0$ | -1.205 | -1.469 | -1.732 | -1.860 | -2.413 | -3.234 | -4.290 | -4.346 | -5.569 |
| $H_{\text{i.h.}}[\varphi_j(x)]/L,\ \varphi_0 = \pi/\beta$ | -0.833 | -1.322 | -1.732 | -1.914 | -2.610 | -3.414 | -4.290 | -4.337 | -5.404 |

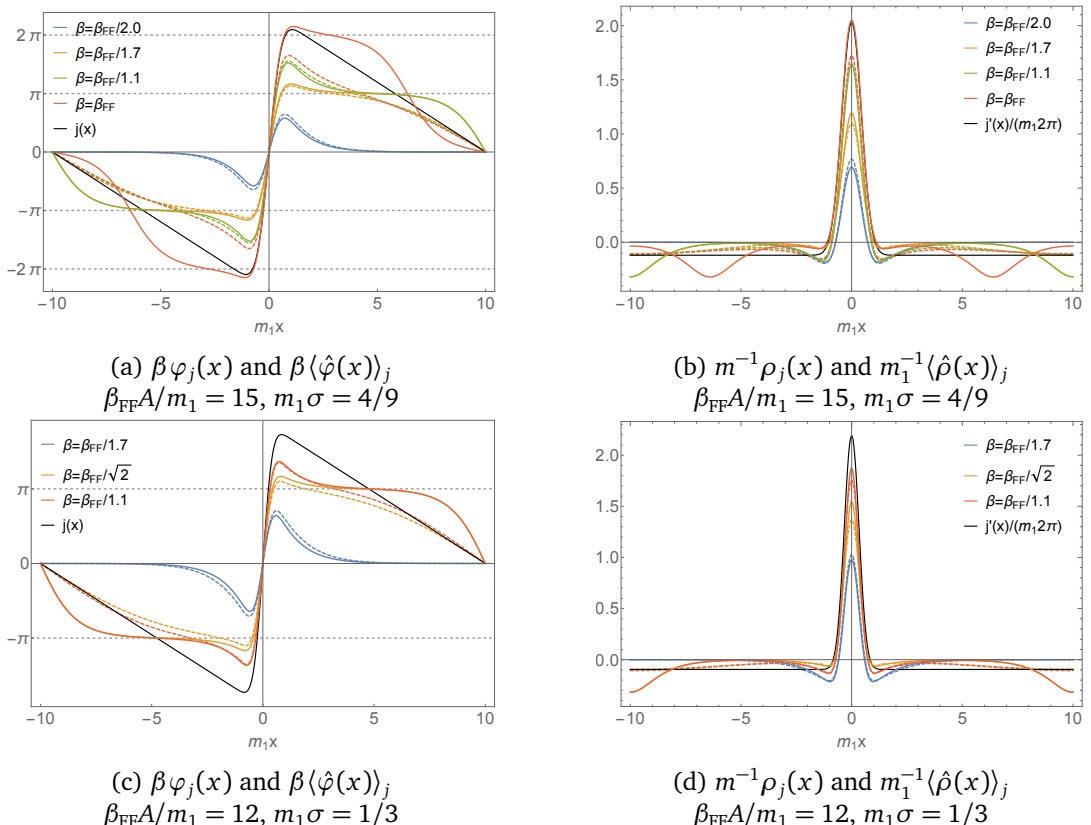

(a) $\beta \varphi_j(x)$ and $\beta \langle \hat{\varphi}(x) \rangle_j$
$\beta_{\mathrm{FF}} A / m_1 = 15$, $m_1 \sigma = 4/9$

(b) $m^{-1} \rho_j(x)$ and $m_1^{-1} \langle \hat{\rho}(x) \rangle_j$
$\beta_{\mathrm{FF}} A / m_1 = 15$, $m_1 \sigma = 4/9$

(c) $\beta \varphi_j(x)$ and $\beta \langle \hat{\varphi}(x) \rangle_j$
$\beta_{\mathrm{FF}} A / m_1 = 12$, $m_1 \sigma = 1/3$

(d) $m^{-1} \rho_j(x)$ and $m_1^{-1} \langle \hat{\rho}(x) \rangle_j$
$\beta_{\mathrm{FF}} A / m_1 = 12$, $m_1 \sigma = 1/3$

Figure B.2: Comparing quantum and classical profiles for different couplings $\beta$ and for two different external field profiles corresponding to different choices of their amplitude paramater $\beta_{\mathrm{FF}} A / m_1$ and width parameter $m_1 \sigma$, as indicated in the corresponding captions.

*Left panel*: The QFT expectation values $\langle \beta \hat{\varphi}(x) \rangle_j$ (dashed lines) and the classical lowest energy configurations $\beta \varphi_j(x)$ (continuous lines).

*Right panel*: The corresponding topological charge densities $m_1^{-1} \langle \hat{\rho}(x) \rangle_j$ in the quantum case (dashed lines) and $m^{-1} \rho_j(x)$ in the classical case (continuous lines). The parameters are $\ell = m_1 L = 20$, $m = m_1$ and different $\beta$ values are shown with different colours and $\beta_{\mathrm{FF}} = \sqrt{4\pi}$. The TCSA profiles $\langle \hat{\rho}(x) \rangle_j$ were extrapolated using cut-offs $e_c = 24, 26, 28$ and 30, and the corresponding profiles $\beta \langle \hat{\varphi}(x) \rangle_j$ were obtained by spatial integration of $2\pi \langle \hat{\rho}(x) \rangle_j = \beta \langle \partial_x \hat{\varphi}(x) \rangle_j$, fixing the zero mode by requiring the result to vanish at the origin $x = 0$. For both choices of the external field profile the classical solutions for two $\beta$ values have $\beta \varphi(0) = \pi$ and are shown shifted by $-\pi$.

Table B.4: The zero mode in the quantum inhomogeneous state, from TCSA, $e_c = 30$, $m_1 L = 20$. $\langle \mathcal{O} \rangle_j$ denotes the expectation value of operator $\mathcal{O}$ in the inhomogeneous initial state, while $\sigma\left[\langle \mathcal{O} \rangle_j\right]$ is the standard deviation characterising the fluctuations. For $R = 1.5$, the $e_c = 28$ ground state and first excited state (V2) are shown as well, since the $e_c = 30$ ground state displays the effects of hybridisation (c.f. Appendix B.1 for more explanation).

| $A\beta_{FF}/m_1 = 12$ $m_1\sigma = 1/3$ | $\langle \cos \beta \hat{\varphi}_0 \rangle_j$ | $\langle \cos^2 \beta \hat{\varphi}_0 \rangle_j$ | $\sigma\left[\langle \cos \beta \hat{\varphi}_0 \rangle_j\right]$ | $\langle \sin \beta \hat{\varphi}_0 \rangle_j$ | $\langle \sin^2 \beta \hat{\varphi}_0 \rangle_j$ | $\sigma\left[\langle \sin \beta \hat{\varphi}_0 \rangle_j\right]$ | $\beta \varphi_0$ |
|---|---|---|---|---|---|---|---|
| $\beta = \beta_{FF}/2.3$ | 0.9667 | 0.9367 | 0.0476 | 0 | 0.0626 | 0.2502 | 0 |
| $\beta = \beta_{FF}/2.0$ | 0.9540 | 0.9144 | 0.0650 | 0 | 0.0846 | 0.2909 | 0 |
| $\beta = \beta_{FF}/1.7$ | 0.9235 | 0.8653 | 0.1111 | 0 | 0.1331 | 0.3648 | 0 |
| $\beta = \beta_{FF}/\sqrt{5/2}$ | 0.8722 | 0.8021 | 0.2036 | 0 | 0.1959 | 0.4426 | $\pi$ |
| $\beta = \beta_{FF}/1.5$ $e_c = 30$ | 0.2914 | 0.5858 | 0.7078 | 0 | 0.4113 | 0.6413 | $\pi$ |
| $\beta = \beta_{FF}/1.5$ $e_c = 28$ GS | 0.7180 | 0.7026 | 0.4325 | 0 | 0.2945 | 0.5427 | $\pi$ |
| $\beta = \beta_{FF}/1.5$ $e_c = 28$ V2 | -0.5147 | 0.6163 | 0.5928 | 0 | 0.3794 | 0.6160 | $\pi$ |
| $\beta = \beta_{FF}/\sqrt{2}$ | -0.4374 | 0.5157 | 0.5696 | 0 | 0.4806 | 0.6933 | $\pi$ |
| $\beta = \beta_{FF}/1.1$ | -0.4514 | 0.5066 | 0.5503 | 0 | 0.4870 | 0.6979 | $\pi$ |
| $\beta = \beta_{FF}$ | -0.4060 | 0.4980 | 0.5772 | 0 | 0.4939 | 0.7028 | $\pi$ |

## C  Free field dynamics

### C.1  Klein–Gordon theory: initial state and dynamics

For the Klein–Gordon theory it is easy write down the inhomogeneous initial state exactly in the eigenbasis of the post-quench Hamiltonian. The analog of the sine–Gordon inhomogeneous Hamiltonian of Eq. (2.16) in the main text reads as

$$
\begin{aligned}
\hat{H}_{\text{inhom}} =&\hat{H}_{\text{KG}} + \hat{H}_j \\
=& \int_{-L/2}^{L/2} dx \left\{ \frac{1}{2}\hat{\pi}(x,t)^2 + \frac{1}{2}(\partial_x \hat{\varphi}(x,t))^2 + \frac{m^2}{2}\hat{\varphi}(x,t)^2 \right\} - \int_{-L/2}^{L/2} dx\, \partial_x \hat{\varphi}(x,t)\, j'(x),
\end{aligned}
\tag{C.1}
$$

and its ground state can be obtained using Bogolyubov transformation.

However, as long as we focus on the expectation value of the field $\hat{\varphi}$, its spatial derivative and their time evolved counterparts, the quantum and classical problems are completely identical. The initial state, or initial profile for $\langle \hat{\varphi} \rangle_j$ and $\langle \partial_x \hat{\varphi} \rangle_j$ can therefore be obtained by the $\beta \to 0$ limit of Eq. (2.19) in the main text

$$
\varphi''(x) = m^2 \varphi(x) + j''(x),
\tag{C.2}
$$

with boundary conditions

$$
\varphi_{\text{odd}}(-L/2) = \varphi_{\text{odd}}(L/2) = 0, \qquad \varphi'_{\text{even}}(-L/2) = \varphi'_{\text{even}}(L/2) = 0.
\tag{C.3}
$$

Recalling that the external field $j(x)$ is an odd function and it admits the standard Fourier representation

$$
j(x) = \sum_{n=-\infty}^{\infty} e^{-i\frac{2\pi n}{L}x} j_n, \qquad j_n = \frac{1}{L}\int_{-L/2}^{L/2} dx\, e^{i\frac{2\pi n}{L}x} j(x),
\tag{C.4}
$$

the expectation value of $\hat{\varphi}$ in the inhomogeneous initial state is

$$\langle \hat{\varphi}(x) \rangle_j = \sum_{n=-\infty}^{\infty} e^{-i\frac{2\pi n}{L}x} j_n \left( \frac{2\pi n}{\omega_n L} \right)^2 , \tag{C.5}$$

where $\omega_n = \sqrt{m^2 + (2\pi n/L)^2}$, $\omega_n = E_{k_n}$ with $k = 2\pi n/L$ and $\langle \hat{\pi} \rangle_j = \langle \partial_t \hat{\varphi} \rangle_j = 0$. Finally, it is easy to check that the above solution is the lowest energy configuration, corresponding to the quantum ground state in the presence of the external source.

For the quantum dynamics it is again sufficient to consider the classical equation of motion

$$\partial_t^2 \varphi = \partial_x^2 \varphi - m^2 \varphi . \tag{C.6}$$

The general solution for arbitrary initial conditions is

$$\varphi(x,t) = \sum_{i=0,1} \int dx' \, G_i(x - x', t) \varphi_i(x'), \tag{C.7}$$

where $\varphi_0 = \varphi(x), \varphi_1 = \pi(x)$ and $G_i(x,t)$ are the corresponding Green's functions. In our particular case with $\varphi(x,0)$ given by (C.5), $\partial_t \varphi(x,0) = \pi(x,0) = 0$ and with periodic boundary conditions, the corresponding solution of the classical equation of motion as well as the evolution of QFT expectation value $\langle \hat{\varphi}(x,t) \rangle_j$ is expressed as

$$\langle \hat{\varphi}(x,t) \rangle_j = \sum_{n=-\infty}^{\infty} e^{-i\frac{2\pi n}{L}x} j_n \left( \frac{2\pi n}{\omega_n L} \right)^2 \cos(\omega_n t), \tag{C.8}$$

from which $\langle \partial_x \hat{\varphi}(x,t) \rangle_j$ is straightforward to obtain.

## C.2 Free fermion Hamiltonian

The other non-interacting limit of the sine–Gordon model requires a full quantum treatment. Using the boson-fermion correspondence it is well known that the homogeneous sine–Gordon Hamiltonian can be rewritten as a massive Dirac Hamiltonian. From bosonisation, it also follows that

$$\rho(x) = \frac{\beta}{2\pi} \partial_x \varphi = \sum_{\sigma=\pm} :\psi_\sigma^\dagger \psi_\sigma:, \tag{C.9}$$

where $\psi$ is a two component spinor field

$$\psi = \begin{pmatrix} \psi_+ \\ \psi_- \end{pmatrix}. \tag{C.10}$$

At the free fermion point $\beta^2 = 4\pi$, the full inhomogeneous Hamiltonian is

$$\begin{aligned} \hat{H}_{\text{inhom}} &= \hat{H}_{\text{D}} + \hat{H}_J \\ &= \int_{-L/2}^{L/2} dx \, \bar{\psi} \left( -i\gamma^1 \partial_1 - M \right) \psi - \int_{-L/2}^{L/2} dx \, J(x) \psi^\dagger(x) \psi(x), \end{aligned} \tag{C.11}$$

where $J(x) = 2\pi/\beta \, j'(x)$ and $M$ is the fermion mass. The gamma matrices in the Weyl representation are

$$\gamma^0 = \begin{pmatrix} 0 & 1 \\ 1 & 0 \end{pmatrix}, \qquad \gamma^1 = \begin{pmatrix} 0 & 1 \\ -1 & 0 \end{pmatrix}, \tag{C.12}$$

and $\bar{\psi} = \psi^\dagger \gamma^0$. The field $\psi$ obeys anti-periodic boundary conditions, $\psi(x + L) = -\psi(x)$, and satisfies the canonical anticommutation relations

$$\{\psi_\sigma(x), \psi_{\sigma'}(y)\} = 0 = \{\psi_\sigma^\dagger(x), \psi_{\sigma'}^\dagger(y)\} \tag{C.13}$$

and

$$\{\psi_\sigma(x), \psi_{\sigma'}^\dagger(y)\} = \delta_{\sigma\sigma'}\delta(x - y). \tag{C.14}$$

It admits the mode expansion

$$\psi(x) = \frac{1}{\sqrt{L}} \sum_n \frac{1}{\sqrt{2E_{p_n}}} \left( a_{p_n} u(p_n) e^{ip_n x} + b_{p_n}^\dagger v(p_n) e^{-ip_n x} \right), \tag{C.15}$$

where $E_k = \sqrt{M^2 + k^2}$, $k$ is quantised in finite volume as $k_n = 2\pi(n + 1/2)/L$ and the mode operators $a$ and $b$ satisfy

$$\{a_p, a_{p'}\} = \{a_p, b_{p'}\} = \{b_p, b_{p'}\} = 0, \tag{C.16}$$

$$\{a_p, b_{p'}^\dagger\} = \{a_p^\dagger, b_{p'}\} = 0, \tag{C.17}$$

$$\{a_p, a_{p'}^\dagger\} = \delta_{pp'} = \{b_p, b_{p'}^\dagger\}. \tag{C.18}$$

The spinor amplitudes $u$ and $v$ are specified as

$$u(p) = \begin{pmatrix} u_+ \\ u_- \end{pmatrix} = \frac{1}{\sqrt{2(E_p + M)}} \begin{pmatrix} E_p + M - p \\ E_p + M + p \end{pmatrix} \tag{C.19}$$

and

$$v(p) = \begin{pmatrix} v_+ \\ v_- \end{pmatrix} = \frac{1}{\sqrt{2(E_p + M)}} \begin{pmatrix} E_p + M - p \\ -(E_p + M) - p \end{pmatrix}. \tag{C.20}$$

The full Hamiltonian in terms of the fermionic creation and annihilation operators is

$$\begin{aligned}
\hat{H}_{\mathrm{D}} + \hat{H}_J = &\sum_n E_{p_n} \left( a_{p_n}^\dagger a_{p_n} + b_{p_n}^\dagger b_{p_n} \right) + \\
&\sum_{n,n'} \frac{1}{2\sqrt{E_{p_n} E_{p_{n'}}}} \Big[ f(p_n, p_{n'}) \left( J_{n'-n} a_{p_n}^\dagger a_{p_{n'}} - J_{n-n'} b_{p_{n'}}^\dagger b_{p_n} \right) \\
&\qquad\qquad + g(p_n, p_{n'}) \left( J_{-n-n'} a_{p_n}^\dagger b_{p_{n'}}^\dagger + J_{n+n'} b_{p_n} a_{p_{n'}} \right) \Big],
\end{aligned} \tag{C.21}$$

where $J_n$ are the Fourier components of the external field $J(x)$ and

$$f(p, p') = \frac{pp' + (M + E_p)(M + E_{p'})}{\sqrt{(M + E_p)(M + E_{p'})}},$$

$$g(p, p') = -\frac{p(M + E_{p'}) + p'(M + E_p)}{\sqrt{(M + E_p)(M + E_{p'})}}.$$

## C.3 Diagonalisation of the inhomogeneous Hamiltonian

Since the Hamiltonian is quadratic in the fermionic creation and annihilation operators, it can be diagonalised using the method of Lieb, Schultz and Mattis [107]. We truncate the set of momenta to $\{p_{-N}, p_{-N+1}, \ldots, p_{N-1}\}$ for some integer $N$. The Hamiltonian can be written in the general form

$$H = \sum_{i,j} c_i^\dagger A_{ij} c_j + \frac{1}{2} c_i^\dagger B_{ij} c_j^\dagger - \frac{1}{2} c_i B_{ij} c_j, \tag{C.22}$$

where $\underline{c}$ is a vector containing the fermionic annihilation operators,

$$
\underline{c} = \begin{pmatrix} a_{p_{-N}} \\ a_{p_{-N+1}} \\ \vdots \\ \hline b_{p_{-N}} \\ b_{p_{-N+1}} \\ \vdots \end{pmatrix},
\tag{C.23}
$$

and the block matrices $\mathbf{A}$ and $\mathbf{B}$ are

$$
A_{ik} = A_{ki}^* = \left( \begin{array}{c|c} \delta_{i,k} E(p_i) - \tilde{f}(p_i, p_k) j_{k-i} & 0 \\ \hline 0 & \delta_{i,k} E(p_i) + \tilde{f}(p_i, p_k) j_{k-i} \end{array} \right)
\tag{C.24}
$$

and

$$
B_{ik} = -B_{ki} = \left( \begin{array}{c|c} 0 & -\tilde{g}(p_i, p_k) j_{-k-i} \\ \hline \tilde{g}(p_i, p_k) j_{-k-i} & 0 \end{array} \right),
\tag{C.25}
$$

where we introduced the shorthand notation

$$
\tilde{f}(p, p') = \frac{f(p, p')}{2\sqrt{E_p E_{p'}}}, \qquad \tilde{g}(p, p') = \frac{g(p, p')}{2\sqrt{E_p E_{p'}}}.
\tag{C.26}
$$

The Hamiltonian (C.22) contains anomalous terms, but it can be brought to the canonical form

$$
H = \sum_k \Lambda_k \eta_k^\dagger \eta_k + \text{const.}
\tag{C.27}
$$

in terms of the new fermionic operators expressed as

$$
\begin{aligned}
\eta_k &= \sum_i (g_{ki} c_i + h_{ki} c_i^\dagger), \\
\eta_k^\dagger &= \sum_i (g_{ki} c_i^\dagger + h_{ki} c_i),
\end{aligned}
\tag{C.28}
$$

with real coefficients. In an obvious matrix-vector notation,

$$
\begin{aligned}
\boldsymbol{\eta} &= \mathbf{G}\boldsymbol{c} + \mathbf{H}\boldsymbol{c}^\dagger, \\
\boldsymbol{\eta}^\dagger &= \mathbf{G}\boldsymbol{c}^\dagger + \mathbf{H}\boldsymbol{c}.
\end{aligned}
\tag{C.29}
$$

The new operators must obey the fermionic canonical commutation relations, so

$$
\begin{aligned}
\{\eta_k, \eta_{k'}^\dagger\} &= \sum_{i,j} \left( g_{ki} g_{k'j} \{c_i, c_j^\dagger\} + h_{ki} h_{k'j} \{c_i^\dagger, c_j\} \right) = \sum_i (g_{ki} g_{k'i} + h_{ki} h_{k'i}) = \delta_{k,k'}, \\
\{\eta_k, \eta_{k'}\} &= \sum_{i,j} \left( g_{ki} h_{k'j} \{c_i, c_j^\dagger\} + h_{ki} g_{k'j} \{c_i^\dagger, c_j\} \right) = \sum_i (g_{ki} h_{k'i} + h_{ki} g_{k'i}) = 0,
\end{aligned}
\tag{C.30}
$$

or

$$
\begin{aligned}
\mathbf{G}\mathbf{G}^{\mathrm{T}} + \mathbf{H}\mathbf{H}^{\mathrm{T}} &= \mathbb{1}, \\
\mathbf{G}\mathbf{H}^{\mathrm{T}} + \mathbf{H}\mathbf{G}^{\mathrm{T}} &= 0.
\end{aligned}
\tag{C.31}
$$

In order to fix the real coefficients, let us compute the commutator $[\eta_k, H]$. By a straightforward calculation and exploiting the symmetry properties of $\mathbf{A}$ and $\mathbf{B}$,

$$
[\eta_k, H] = \sum_{i,j} [(A_{ji} g_{kj} - B_{ji} h_{kj}) c_i + (B_{ji} g_{kj} - A_{ij} h_{kj}) c_i^\dagger].
\tag{C.32}
$$

Using Eq. (C.27) this should be equal to

$$[\eta_k, H] = \Lambda_k \eta_k = \Lambda_k \sum_i (g_{ki} c_i + h_{ki} c_i^\dagger), \tag{C.33}$$

which implies

$$\Lambda_k g_{ki} = \sum_j (g_{kj} A_{ji} - h_{kj} B_{ji}),$$
$$\Lambda_k h_{ki} = \sum_j (g_{kj} B_{ji} - h_{kj} A_{ij}). \tag{C.34}$$

Introducing the matrix $\mathbf{\Lambda}$ such that $\mathbf{\Lambda}_{ik} = \Lambda_i \delta_{i,k}$, we can write this as

$$\mathbf{\Lambda G} = \mathbf{GA} - \mathbf{HB},$$
$$\mathbf{\Lambda H} = \mathbf{GB} - \mathbf{HA}, \tag{C.35}$$

To decouple these equations, we introduce

$$\mathbf{\Phi} = \mathbf{G} + \mathbf{H},$$
$$\mathbf{\Psi} = \mathbf{G} - \mathbf{H}, \tag{C.36}$$

in terms of which

$$\mathbf{\Lambda \Phi} = \mathbf{\Psi}(\mathbf{A} + \mathbf{B}),$$
$$\mathbf{\Lambda \Psi} = \mathbf{\Phi}(\mathbf{A} - \mathbf{B}). \tag{C.37}$$

Multiplying by $\mathbf{A} - \mathbf{B}$ and $\mathbf{A} + \mathbf{B}$ we obtain

$$\mathbf{\Lambda}^2 \mathbf{\Phi} = \mathbf{\Phi}(\mathbf{A} - \mathbf{B})(\mathbf{A} + \mathbf{B}),$$
$$\mathbf{\Lambda}^2 \mathbf{\Psi} = \mathbf{\Phi}(\mathbf{A} + \mathbf{B})(\mathbf{A} - \mathbf{B}). \tag{C.38}$$

These are eigenvalue equations of the Hermitian matrices $(\mathbf{A} - \mathbf{B})(\mathbf{A} + \mathbf{B})$ and $(\mathbf{A} + \mathbf{B})(\mathbf{A} - \mathbf{B})$, where the common eigenvalues are $\Lambda_k^2$ and the eigenvectors $\underline{\Phi}_k$ and $\underline{\Psi}_k$ are the rows of the matrices $\mathbf{\Phi}$ and $\mathbf{\Psi}$ (i.e. $(\underline{\Phi}_k)_i = \Phi_{ki}, (\underline{\Psi}_k)_i = \Psi_{ki}$). Each set of eigenvectors forms a complete orthonormal basis. Orthonormality is equivalent to Eqs. (C.31):

$$(\mathbf{G} + \mathbf{H})(\mathbf{G} + \mathbf{H})^T = \mathbb{1} \Longleftrightarrow \underline{\Phi}_k \cdot \underline{\Phi}_{k'} = \delta_{k,k'},$$
$$(\mathbf{G} - \mathbf{H})(\mathbf{G} - \mathbf{H})^T = \mathbb{1} \Longrightarrow \underline{\Psi}_k \cdot \underline{\Psi}_{k'} = \delta_{k,k'}. \tag{C.39}$$

In a similar fashion, completeness can be written as

$$(\mathbf{G} + \mathbf{H})^T(\mathbf{G} + \mathbf{H}) = \mathbb{1} \Longrightarrow \sum_k \Phi_{ki} \Phi_{kj} = \delta_{i,j},$$
$$(\mathbf{G} - \mathbf{H})^T(\mathbf{G} - \mathbf{H}) = \mathbb{1} \Longleftrightarrow \sum_k \Psi_{ki} \Psi_{kj} = \delta_{i,j}. \tag{C.40}$$

These relations allow us to invert the transformation (C.29):

$$\mathbf{c} = \mathbf{G}^T \boldsymbol{\eta} + \mathbf{H}^T \boldsymbol{\eta}^\dagger,$$
$$\mathbf{c}^\dagger = \mathbf{G}^T \boldsymbol{\eta}^\dagger + \mathbf{H}^T \boldsymbol{\eta}. \tag{C.41}$$

Solving the eigenvalue problems yield the matrices $\mathbf{\Phi}$ and $\mathbf{\Psi}$ from which the transformation matrices $\mathbf{G}$ and $\mathbf{H}$ can be obtained. The ground state is given by the Fermi sea in which all the

modes with $\Lambda_k < 0$ are filled. It is convenient to perform a particle-hole transformation for these modes such that the ground state will be the empty state. Indeed, the sign of $\Lambda_k$ can be flipped by the transformations

$$\boldsymbol{\Phi} \to -\boldsymbol{\Phi}, \boldsymbol{\Psi} \to \quad \boldsymbol{\Psi} \quad \Longleftrightarrow \quad \mathbf{G} \to -\mathbf{H}, \mathbf{H} \to -\mathbf{G} \quad \Longleftrightarrow \quad \boldsymbol{\eta} \leftrightarrow -\boldsymbol{\eta}^\dagger, \tag{C.42}$$

$$\boldsymbol{\Phi} \to \quad \boldsymbol{\Phi}, \boldsymbol{\Psi} \to -\boldsymbol{\Psi} \quad \Longleftrightarrow \quad \mathbf{G} \to \quad \mathbf{H}, \mathbf{H} \to \quad \mathbf{G} \quad \Longleftrightarrow \quad \boldsymbol{\eta} \leftrightarrow \quad \boldsymbol{\eta}^\dagger, \tag{C.43}$$

which are particle-hole transformations in terms of the modes $\eta_k$. In practice, this particle-hole transformation is implemented by determining the sign of $\Lambda_k$ from

$$S_k = \mathrm{sgn}(\Lambda_k) = \mathrm{sgn}\left(\underline{\Psi}_k^T (\mathbf{A} + \mathbf{B}) \underline{\Phi}_k\right) = \mathrm{sgn}\left(\underline{\Phi}_k^T (\mathbf{A} - \mathbf{B}) \underline{\Psi}_k\right), \tag{C.44}$$

and then defining the modes via

$$\mathbf{G} = (\boldsymbol{\Phi} + \mathbf{S}\boldsymbol{\Psi})/2,$$
$$\mathbf{H} = (\boldsymbol{\Phi} - \mathbf{S}\boldsymbol{\Psi})/2, \tag{C.45}$$

where $\mathbf{S}_{ik} = S_k \delta_{ik}$. In terms of these modes the ground state satisfies

$$\eta_k |0\rangle = 0. \tag{C.46}$$

## C.4 Dynamics of Dirac charge density

The (normal ordered) local charge density operator is given by

$$\rho(x) = \psi^\dagger(x)\psi(x) = \frac{1}{L} \sum_{p,p'} \frac{1}{2\sqrt{E_p E_{p'}}} \cdot \Big[ a_p^\dagger a_{p'} f(p,p') e^{-i(p-p')x} + a_p^\dagger b_{p'}^\dagger g(p,p') e^{-i(p+p')x}$$

$$- a_p b_{p'} g(p,p') e^{i(p+p')x} - b_{p'}^\dagger b_p f(p,p') e^{i(p-p')x} \Big]$$

$$= \sum_{i,j} c_i^\dagger D_{ij} c_j + \frac{1}{2} c_i^\dagger E_{ij} c_j^\dagger - \frac{1}{2} c_i E_{ij}^* c_j = \mathbf{c}^\dagger \mathbf{D} \mathbf{c} + \frac{1}{2} (\mathbf{c}^\dagger \mathbf{E} \mathbf{c}^\dagger - \mathbf{c} \mathbf{E}^* \mathbf{c}), \tag{C.47}$$

where the matrices are

$$D_{ik} = D_{ki}^* = \frac{1}{L} \left( \begin{array}{c|c} \tilde{f}(p_i, p_k) e^{-i(p_i - p_k)x} & 0 \\ \hline 0 & -\tilde{f}(p_i, p_k) e^{-i(p_i - p_k)x} \end{array} \right) \tag{C.48}$$

and

$$E_{ik} = -E_{ki} = \frac{1}{L} \left( \begin{array}{c|c} 0 & \tilde{g}(p_i, p_k) e^{-i(p_i + p_k)x} \\ \hline -\tilde{g}(p_i, p_k) e^{-i(p_i + p_k)x} & 0 \end{array} \right). \tag{C.49}$$

Rewriting it in terms of the $\eta$-fermions,

$$\rho(x) = \boldsymbol{\eta}^T \mathbf{H} \mathbf{D} \mathbf{H}^T \boldsymbol{\eta}^\dagger + \frac{1}{2} \boldsymbol{\eta} \mathbf{H} \mathbf{E} \mathbf{G}^T \boldsymbol{\eta}^\dagger - \frac{1}{2} \boldsymbol{\eta} \mathbf{G} \mathbf{E}^* \mathbf{H}^T \boldsymbol{\eta}^\dagger + [9 \text{ normal ordered terms}]. \tag{C.50}$$

Taking the vacuum expectation value, the diagonal elements of the matrices will survive, and the sum over them yields the trace:

$$\rho(x) = \mathrm{Tr}\left( \mathbf{H} \mathbf{D} \mathbf{H}^T + \frac{1}{2} \mathbf{H} \mathbf{E} \mathbf{G}^T - \frac{1}{2} \mathbf{G} \mathbf{E}^* \mathbf{H}^T \right). \tag{C.51}$$

The time evolution under the free Hamiltonian $H_0$ is easy to obtain. In the Heisenberg picture,

$$\mathbf{c}(t) = \begin{pmatrix} a_{p_{-N}} e^{-iE_{p_{-N}} t} \\ \vdots \\ b_{p_{-N}} e^{-iE_{p_{-N}} t} \\ \vdots \end{pmatrix} = \mathbf{c} \mathbf{U}, \tag{C.52}$$

where

$$\mathbf{U} = \begin{pmatrix} \begin{matrix} e^{-iE_{p-N}t} & 0 \\ 0 & \ddots \end{matrix} & \\ & \begin{matrix} e^{-iE_{p-N}t} & 0 \\ 0 & \ddots \end{matrix} \end{pmatrix}. \tag{C.53}$$

Then

$$\rho(x,t) = \mathrm{Tr}\left(\mathbf{H}\mathbf{D}(t)\mathbf{H}^T + \frac{1}{2}\mathbf{H}\mathbf{E}(t)\mathbf{G}^T - \frac{1}{2}\mathbf{G}\mathbf{E}(t)^*\mathbf{H}^T\right), \tag{C.54}$$

where $\mathbf{D}(t) = \mathbf{U}^\dagger \mathbf{D}\mathbf{U}$ and $\mathbf{E}(t) = \mathbf{U}^\dagger \mathbf{E}\mathbf{U}$. The total charge is then simply

$$Q = \int_{-L/2}^{L/2} \mathrm{d}x\,\rho(x,t) = \mathrm{Tr}\left(\mathbf{H}\mathbf{H}^T\right). \tag{C.55}$$

## C.5 Derivation of the LDA formula

Although use of the LDA and in particular the formula Eq. (4.3) of the main text is very common in the literature, we find it instructive to present a short derivation because it highlights the underlying assumptions of LDA that are responsible for its failure for external inhomogeneities of small amplitude.

Let us recall that the inhomogeneous Hamiltonian has the form

$$\hat{H}_{\mathrm{inhom}} = \int \mathrm{d}x\,\bar{\psi}\left(-i\gamma^1\partial_1 - M\right)\psi - \int \mathrm{d}x\,\mu(x)\psi^\dagger(x)\psi(x). \tag{C.56}$$

We define the particle number operator

$$\hat{N} = \int \mathrm{d}x\,\psi^\dagger(x)\psi(x), \tag{C.57}$$

which counts the number of fermions minus the number of antifermions. This operator commutes with both the homogeneous part of the Hamiltonian (C.56) as well as with its non-homogeneous term for constant chemical potentials $\mu$.

The first approximation entering in LDA is that we assume that the system can be divided into subsystems restricted to finite boxes, in which $\mu(x)$ can be regarded as a constant. Denoting the size of such a box by $L$ and the chemical potential therein by $\mu$, we can write

$$\hat{H}_{\mathrm{inhom}} = \int_{L/2}^{L/2} \mathrm{d}x\,\bar{\psi}\left(-i\gamma^1\partial_1 - M\right)\psi - \mu\int_{L/2}^{L/2} \mathrm{d}x\,\psi^\dagger(x)\psi(x), \tag{C.58}$$

for one box.

The box must be larger than the microscopic scales which in our case are set by the Compton wavelength (or correlation length) $M^{-1}$. This also ensures that we can disregard the boundary conditions and possible boundary energies. The density and thus the chemical potential must vary slowly on the scale of $L$ to be able to neglect the energy cost of derivative terms, so $|\mu(x)/\mu'(x)| \gg L$, which together with $L \gg M^{-1}$ implies $M \gg |\mu'(x)/\mu(x)|$.

The commutation of $\hat{N}$ and $\hat{H}_{\mathrm{inhom}}$ with constant $\mu$ straightforwardly implies that the ground state (as well as other eigenstates) can be indexed by the quantum number $N$, that is by the eigenvalues of $\hat{N}$. Thus, we can easily find the average density $\langle\hat{\rho}\rangle$ in this fluid cell by using energetic arguments and exploit that the ground state of the box Hamiltonian has a definite particle number $N$. In particular, to find the ground state energy and density, we have to minimise the energy of (C.58) with respect to $N$. Here we note that to minimise the

positive energy contribution of the homogeneous Hamiltonian, we can choose either the number of fermions or that of the antifermions to be zero, and distribute the $N$ particles evenly in momentum space around zero momentum.

We can thus write

$$E = \sum_{n=-N/2}^{N/2} E_{p_n} - \mu N\,, \tag{C.59}$$

where $E_{p_n}$ is the standard relativistic dispersion relation $\sqrt{M^2 + (2\pi n/L)^2}$ and the momentum quantisation corresponds to periodic boundary conditions. If besides the condition $M \gg |\frac{d\mu(x)}{dx}/\mu(x)|$ one additionally assumes that $N \gg 1$ in the fluid cell, then the sum can be replaced by an integral

$$E = \frac{L}{2\pi} \int_{-2\pi N/(2L)}^{2\pi N/(2L)} dn \sqrt{M^2 + n^2} - \mu N\,. \tag{C.60}$$

This new assumption implicitly requires that the magnitude of the local chemical potential cannot be too small, since a mesoscopic number of particles has to be excited. We can now easily differentiate the above integral with respect to $N$ to find the minimum and the optimal number of excitations. This yields the expected result

$$2\sqrt{M^2 + \left(\frac{2\pi N}{2L}\right)^2} - \mu = 0\,, \tag{C.61}$$

which corresponds to filling the Dirac sea up to the chemical potential by positive and negative momentum particles. Now the local density $\langle \hat{\rho} \rangle = N/L$ can be immediately expressed as

$$\langle \hat{\rho} \rangle = \frac{1}{\pi} \sqrt{\left(\frac{\mu}{2}\right)^2 - M^2}\,. \tag{C.62}$$

Considering more carefully the sign and the real valued nature of the solution one straightforwardly obtains

$$\langle \hat{\rho} \rangle = \frac{1}{\pi} \begin{cases} \sqrt{(\mu/2)^2 - M^2} & \text{if } \mu/2 \geq M\,, \\ -\sqrt{(\mu/2)^2 - M^2} & \text{if } \mu/2 \leq -M\,, \\ 0 & \text{otherwise}\,. \end{cases} \tag{C.63}$$

Joining the fluid cells together we end up with

$$\langle \hat{\rho}(x) \rangle = \frac{1}{\pi} \begin{cases} \sqrt{(\mu(x)/2 + \mu_0)^2 - M^2} & \text{if } \mu(x)/2 + \mu_0 \geq M\,, \\ -\sqrt{(\mu(x)/2 + \mu_0)^2 - M^2} & \text{if } \mu(x)/2 + \mu_0 \leq -M\,, \\ 0 & \text{otherwise}\,, \end{cases} \tag{C.64}$$

where the global $\mu_0$ is introduced to adjust the total charge in the full system to a prescribed value (e.g. zero).

To summarise, in the derivation of the LDA result Eq. (C.64) we supposed that $\mu(x)$ varies slowly on the scale of the fluid cells. Since the size $L$ of the fluid cells must be greater than the soliton Compton length $M^{-1}$, this can be phrased as $|\mu(x)/\mu'(x)| \gg M^{-1}$. We also had to assume that the number of excitations in a fluid cell is macroscopic, which can be formulated as the requirement $\langle \hat{\rho}(x) \rangle L \gg 1$, implying the condition $\langle \hat{\rho}(x) \rangle |\mu(x)/\mu'(x) \gg 1$. Checking the first condition $|\frac{d\mu(x)}{dx}/\mu(x)| \ll M$ is straightforward, while the second one is equivalent, for sufficiently slowly varying external chemical potentials of high enough amplitude, to $\mu(x)^2/|\frac{d\mu(x)}{dx}| \gg 1$.

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
