# Peer review of "Inhomogeneous quantum quenches in the sine-Gordon theory"

_SciPost Physics, doi:SciPost Phys. 12, 144 (2022)_

## Round 2 · Referee Report · Anonymous (Referee 1) · 2022-1-31

Strengths

  • careful and detailed analysis of particular inhomogeneous quenches in the quantum sine-Gordon model
  • high quality numerical results (using TSCA) for physically relevant observable

Report

The authors study a particular class of inhomogeneous quantum quenches in the attractive regime of the sine-Gordon model (SGM) by means of the Truncated Conformal Space Approach and free fermion methods. The class of initial states is obtained by coupling the SGM to an external source and preparing the resulting system in its ground state. The source is then removed and time evolution of the topological charge density is studied.

The non-equilibrium dynamics in the sine-Gordon model is an important problem of experimental relevance and this work reports interesting progress in several regards. Perhaps the most important message of this work is that inhomogeneous quenches in the SGM can be qualitatively described by TCSA. Secondly, the initial states are shown to exhibit some interesting behavior as a function of the parameter $\beta$ of the SGM ("ground state phase transitions"). While this can be readily understood in the classical limit, the results in the quantum case are interesting and to the best of my knowledge new. Thirdly, the dynamics of the topological charge density after these quenches is quite rich and displays interesting cross-overs between the semiclassical and free fermion limits of the SGM. I recommend publication but would ask the authors to consider the following points:

  1. I think it is essential to add a fully quantitative comparison between TSCA and the exact free fermion result. The color plots in Fig. IV.9 show that TSCA appears to be generally in good agreement with the numerically exact result. However, I think readers will want to have a much more quantitative comparison. A simple way of providing this would be to show representative "cuts" for fixed x/L as a function of time.

  2. The authors stress the failure of the local density approximation. As they discuss, it is well known that LDA has a regime of validity and is not expected to work outside it. The criteria for LDA to hold in the free fermion case are given in Appendix C.5 and I think it would be useful to phrase the discussion in IV.B. in terms of these criteria. Perhaps the authors will want to comment about using the LDA in combination with the Bethe Ansatz solution away from the free fermion point.

  3. A natural question is whether there are any easily identifiable features associated with the first breather in the weakly attractive regime. It seems that the answer is no (as no such feature is mentioned), but I think it would be helpful for the readers if this was spelled out clearly.

  • validity: top
  • significance: high
  • originality: good
  • clarity: high
  • formatting: excellent
  • grammar: excellent

Author:  Márton Kormos  on 2022-03-24  [id 2319]

(in reply to Report 1 on 2022-01-31)
Category:
answer to question

We thank Referee 1 for the valuable remarks. Our replies are the following:

Following the advice of the Referee, we have added an additional figure (IV.12) to demonstrate the comparison between the exact FF dynamics and that of TCSA. In addition, we now also show the extrapolated time evolved quantities to have a better insight into the accuracy and limitations of TCSA. However, we would like to stress again that the use of our extrapolating procedure is not strictly justified for time-evolved quantities (unlike for equilibrium ones), nevertheless we think such quantities may be useful to understand better TCSA in out-of-equilibrium settings.

We have rewritten some parts of Appendix C.5 and we now explicitly quote the two conditions for the applicability of LDA in Sec. IV.B, see Eqs. (IV.4) and (IV.5) and the discussion around them. We refrain from extending our LDA calculations away from the free fermion point as this requires much more sophisticated methods that are out of the scope of the present work; however, we may return to this problem in the near future.

We are grateful for drawing our attention to this point and have added some additional text in Section IV.C (first two paragraphs on p. 17) to better explain the effect of the first breather close to the free-fermion point or in other words in a weakly interacting regime (which is not to be confused with the free-boson point of the theory). We have demonstrated that exactly at the free-fermion point a fermion and an antifermion (soliton-antisoliton pair) form a bosonic collective excitation of mass 2M, where M is the soliton mass. When moving away from this point, the 1st breather, an inherent bosonic excitation indeed enters the spectrum. Nevertheless its mass is almost equal to 2M, and as indicated by the wave function renormalisation constant, its matrix elements with respect to the canonical field (or its derivatives) are small. These factors mean the impact of the 1st breather on observables that are functions of the fundamental field is very small. The disentangling of such a small effect, in principle, can be possible via a proper Fourier analysis of the time evolved quantities. However, due to the minor mass difference between the mass of the 1st breather and soliton-antisoliton excitation, one needs much longer time windows to carry out such an analysis than the ones in which we are able to perform our simulations.

---

## Round 2 · Referee Report · Anonymous (Referee 2) · 2022-2-2

Strengths

1- Detailed discussion and analysis of inhomogeneous quench 2-Number of useful appendices which makes the whole story self-contained

Weaknesses

1- Some formulations could be improved

Report

The authors study quench dynamics of sine-Gordon model with inhomogeneous background of topological charge in the finite-size system. Using truncated conformal space approach they investigate details of subsequent time evolution governed by homogeneous sine-Gordon Hamiltonian. Authors discuss in details some observed features of topological charge profile and expectation values of several observables. They observe an interesting crossover phenomena both in the initial state and in quantum dynamics. The paper is interesting and should be accepted for publication after some minor remarks are implemented.

Requested changes

1- I suggest to clarify a bit more what is meant by oscillatory behaviour. In particular on page 16, last paragraph the authors say: "Indeed, it can be easily seen from Figs. IV.6 and IV.9 that the dominant oscillation frequency approximately equals 2π which is associated with a single boson at rest". I must admit this statement is not very obvious... My understanding of this oscillation is a color change on these plots as we move forward in time (is it correct?). Please clarify this. Related question is why the lighter breathers do not show up in this oscillatory behaviour (of course in the regime of the $\beta$ parameter where they are present)? 2- Can the setup and the results in the paper be modelled by the integrable hydrodynamics approach? If so, what would be the relevant time scale for the integrable hydrodynamics to work? Please comment on this. 3- I think the prefactor in eq. C.60 is wrong (it should be $L/2\pi$ instead, I think), otherwise we do not get eq. C.61 as it is written now (which I think is correct). 4- In the introduction I would recommend to mention commensurate-incommensurate transition (and so the old papers by Haldane, Japaridze and Nersessyan) in the limiting case where profile is constant in space (homogeneous), so that the perturbation is just proportional to the topological charge density.
5-I suggest to go through the text once again to avoid some trivial repetitions of words and to improve some formulations.

  • validity: high
  • significance: high
  • originality: high
  • clarity: high
  • formatting: excellent
  • grammar: good

Author:  Márton Kormos  on 2022-03-24  [id 2320]

(in reply to Report 2 on 2022-02-02)

We thank Referee 2 for the valuable suggestions. Our replies are the following:

  1. We have added an additional figure to better highlight that the period of the oscillations after the initial bump splits equals 2 π. To this end, we plotted the time evolution of the expectation value of the topological charge density at a fixed position x=0 (Fig. IV. 11). The effect of the other breathers is generically less pronounced in various quantities during the time evolution. One way to identify their effect is via Fourier analysis (a.k.a. “quench spectroscopy”), however it requires longer time windows than those we were able to simulate. In addition, these much longer simulation times are also needed to achieve precision sufficient to distinguish numerically the frequencies of higher breathers from those of multi-particle states containing lower breathers.

  2. Integrable hydrodynamic approaches applicable for the sine-Gordon have only been developed for the repulsive regime of the model and for very particular initial states. The quench studied in our work takes place in the attractive regime and it is not known how the particular characterisation of the inhomogeneous initial state that is necessary for integrable hydrodynamical approaches can be carried out. We would like to stress that to the best of our knowledge, the precise conditions of hydrodynamical approaches, such as from what time scales they provide an accurate description of the physical system, have not yet been explored.

  3. We have corrected the prefactor.

  4. We have added the references to the end of paragraph 4 of the Introduction. In addition, we added a reference to a related work by Pokrovsky and Talapov (also mentioned in the Conclusions).

  5. We have gone through the text and fixed a number of such repetitions, and improved formulations.

---

## Round 3 · Author Response

We thank the referees for their constructive suggestions and the questions they raised. Below we answer each point separately and explain the changes we have made to the paper.

Referee 1:

Report:

  1. I think it is essential to add a fully quantitative comparison between TSCA and the exact free fermion result. The color plots in Fig. IV.9 show that TSCA appears to be generally in good agreement with the numerically exact result. However, I think readers will want to have a much more quantitative comparison. A simple way of providing this would be to show representative "cuts" for fixed x/L as a function of time.
  2. The authors stress the failure of the local density approximation. As they discuss, it is well known that LDA has a regime of validity and is not expected to work outside it. The criteria for LDA to hold in the free fermion case are given in Appendix C.5 and I think it would be useful to phrase the discussion in IV.B. in terms of these criteria. Perhaps the authors will want to comment about using the LDA in combination with the Bethe Ansatz solution away from the free fermion point.
  3. A natural question is whether there are any easily identifiable features associated with the first breather in the weakly attractive regime. It seems that the answer is no (as no such feature is mentioned), but I think it would be helpful for the readers if this was spelled out clearly.

We thank Referee 1 for the valuable remarks. Our replies are the following:

  1. Following the advice of the Referee, we have added an additional figure (IV.12) to demonstrate the comparison between the exact FF dynamics and that of TCSA. In addition, we now also show the extrapolated time evolved quantities to have a better insight into the accuracy and limitations of TCSA. However, we would like to stress again that the use of our extrapolating procedure is not strictly justified for time-evolved quantities (unlike for equilibrium ones), nevertheless we think such quantities may be useful to understand better TCSA in out-of-equilibrium settings.

  2. We have rewritten some parts of Appendix C.5 and we now explicitly quote the two conditions for the applicability of LDA in Sec. IV.B, see Eqs. (IV.4) and (IV.5) and the discussion around them. We refrain from extending our LDA calculations away from the free fermion point as this requires much more sophisticated methods that are out of the scope of the present work; however, we may return to this problem in the near future.

  3. We are grateful for drawing our attention to this point and have added some additional text in Section IV.C (first two paragraphs on p. 17) to better explain the effect of the first breather close to the free-fermion point or in other words in a weakly interacting regime (which is not to be confused with the free-boson point of the theory). We have demonstrated that exactly at the free-fermion point a fermion and an antifermion (soliton-antisoliton pair) form a bosonic collective excitation of mass 2M, where M is the soliton mass. When moving away from this point, the 1st breather, an inherent bosonic excitation indeed enters the spectrum. Nevertheless its mass is almost equal to 2M, and as indicated by the wave function renormalisation constant, its matrix elements with respect to the canonical field (or its derivatives) are small. These factors mean the impact of the 1st breather on observables that are functions of the fundamental field is very small. The disentangling of such a small effect, in principle, can be possible via a proper Fourier analysis of the time evolved quantities. However, due to the minor mass difference between the mass of the 1st breather and soliton-antisoliton excitation, one needs much longer time windows to carry out such an analysis than the ones in which we are able to perform our simulations.

Referee 2:

Report: 1- I suggest to clarify a bit more what is meant by oscillatory behaviour. In particular on page 16, last paragraph the authors say: "Indeed, it can be easily seen from Figs. IV.6 and IV.9 that the dominant oscillation frequency approximately equals 2π which is associated with a single boson at rest". I must admit this statement is not very obvious... My understanding of this oscillation is a colour change on these plots as we move forward in time (is it correct?). Please clarify this. Related question is why the lighter breathers do not show up in this oscillatory behaviour (of course in the regime of the β parameter where they are present)? 2- Can the setup and the results in the paper be modelled by the integrable hydrodynamics approach? If so, what would be the relevant time scale for the integrable hydrodynamics to work? Please comment on this. 3- I think the prefactor in eq. C.60 is wrong (it should be L/(2 π) 2 π/L instead, I think), otherwise we do not get eq. C.61 as it is written now (which I think is correct). 4- In the introduction I would recommend to mention commensurate-incommensurate transition (and so the old papers by Haldane, Japaridze and Nersessyan) in the limiting case where profile is constant in space (homogeneous), so that the perturbation is just proportional to the topological charge density. 5-I suggest to go through the text once again to avoid some trivial repetitions of words and to improve some formulations.

We thank Referee 2 for the valuable suggestions. Our replies are the following:

  1. We have added an additional figure to better highlight that the period of the oscillations after the initial bump splits equals 2 π. To this end, we plotted the time evolution of the expectation value of the topological charge density at a fixed position x=0 (Fig. IV. 11).

The effect of the other breathers is generically less pronounced in various quantities during the time evolution. One way to identify their effect is via Fourier analysis (a.k.a. “quench spectroscopy”), however it requires longer time windows than those we were able to simulate. In addition, these much longer simulation times are also needed to achieve precision sufficient to distinguish numerically the frequencies of higher breathers from those of multi-particle states containing lower breathers.

  1. Integrable hydrodynamic approaches applicable for the sine-Gordon have only been developed for the repulsive regime of the model and for very particular initial states. The quench studied in our work takes place in the attractive regime and it is not known how the particular characterisation of the inhomogeneous initial state that is necessary for integrable hydrodynamical approaches can be carried out. We would like to stress that to the best of our knowledge, the precise conditions of hydrodynamical approaches, such as from what time scales they provide an accurate description of the physical system, have not yet been explored.

  2. We have corrected the prefactor.

  3. We added the references to the end of paragraph 4 of the Introduction. In addition, we added a reference to a related work by Pokrovsky and Talapov (also mentioned in the Conclusions).

  4. We have gone through the text and fixed a number of such repetitions, and improved formulations.

---

## Round 3 · List of Changes

• For the changes indicated in our reply to the referee reports see above. These included adding further text, two new figures (Figs. IV.11 and IV.12) and additional references (Ref. [64-68]).
  • We have also added a new reference ([104]), which presents in detail the numerical method used in this work.
  • Lastly, we went through the text and corrected misprints, typos and also made some improvements to the style.

---

## Editorial Decision

published